# Assessing flash flood erosion following storm Daniel in Libya

Jonathan C. L. Normand [1,2] & Essam Heggy [1,3] ✉

The eastern Mediterranean basin is witnessing increased storm activity impacting populous urban coastal areas that historically were not prone to catastrophic flooding. In the fall of 2023, Storm Daniel struck the eastern coast of Libya, causing unprecedented flash floods with a tragic death toll and large-scale infrastructure damages. We use Sentinel-1A C-band SAR images to characterize the resulting flash flood erosion and sediment load dynamics across the watersheds and to map damages within coastal cities at their outlets. Our results suggest that sediment loading, resulting from surface erosion, increased the density of turbid streams. The above exacerbated the catastrophic impact of the flash floods in the coastal cities of Derna and Susah, where 66% and 48% of their respective urban surface have experienced moderate-to-high damages. Our findings highlight the increased vulnerability of coastal watersheds in arid areas within the eastern Mediterranean basin due to the forecasted increase in hydroclimatic extremes and call for a transformative coastal management approach to urgently implement nature-based solutions and land-use changes to mitigate these rising risks.

Severe cyclone-type storms, also known as medicanes, are generated from the combination of a cold cut-off low in the middle to upper-troposphere and warm seawater, naturally occurring during the autumn season[1]. They form in the central Mediterranean basin but typically dissipate at sea[2]. However, these medicanes have increased in duration and magnitude over the last decades and moved toward the low-lying eastern Mediterranean coasts of Egypt and Libya, leading to catastrophic landfall[2]. Over the last decades, the impacts of these hydroclimatic extremes were puzzling due to their sparse occurrence[3] and the lack of in-situ observations as a result of the low population density in these coastal areas. Furthermore, the impacts of these intensifying storms remain poorly quantified as most post-assessment efforts naturally focus on identifying the damages in localized urban zones, overlooking the regional physical drivers that modulate the magnitude of these flash floods. Notably, the vulnerabilities for these low-lying arid sandy coasts were low until the recent sharp increase in local population due to the civil war causing inhabitants displacements and immigration from other African countries[4], as well as the rise in medicanes intensity. The impacts of these medicanes have recently been subject to both public and scientific attention[2,5,6].

In September 2023, the medicane Storm Daniel[7] developed in Greece over the Ionian Sea and strengthened while moving southward across the eastern Mediterranean Sea. The storm was amplified by the exceptionally warm waters resulting from the extreme summer heat in southern Europe that year[8]. Storm Daniel reached the northeastern coast of Libya on September 10th, 2023, with high winds of 120 km h$^{-1}$ and 25 hours of cumulative precipitation reaching 240 mm (up to 414 mm in the city of Al-Bayda). The above-observed precipitation during the storm landfall is close to the annual average of 270 mm yr$^{-1}$ for this area[9]. Although rare, the area is known to experience intermittent extreme rainstorms and flash floods during the fall season. Notably, over the past 80 years, five flash floods have been recorded in Wadi-Derna, a dry riverbed, occurring in October 1942, October 1959, October 1968, November 1986 and September 2011[10].

The extreme hydroclimatic conditions generated by Storm Daniel caused deluges and mudflows in the storm landfall area (see Fig. 1) primarily constituted of rangelands (grasses and shrubs) and bare

[1]University of Southern California, Viterbi School of Engineering, Los Angeles, CA 90089, USA. [2]University Paris Cité, Institut de Physique du Globe de Paris, CNRS, Paris 75005, France. [3]NASA's Jet Propulsion Laboratory, California Institute of Technology, Pasadena, CA 91109, USA. ✉e-mail: heggy@usc.edu

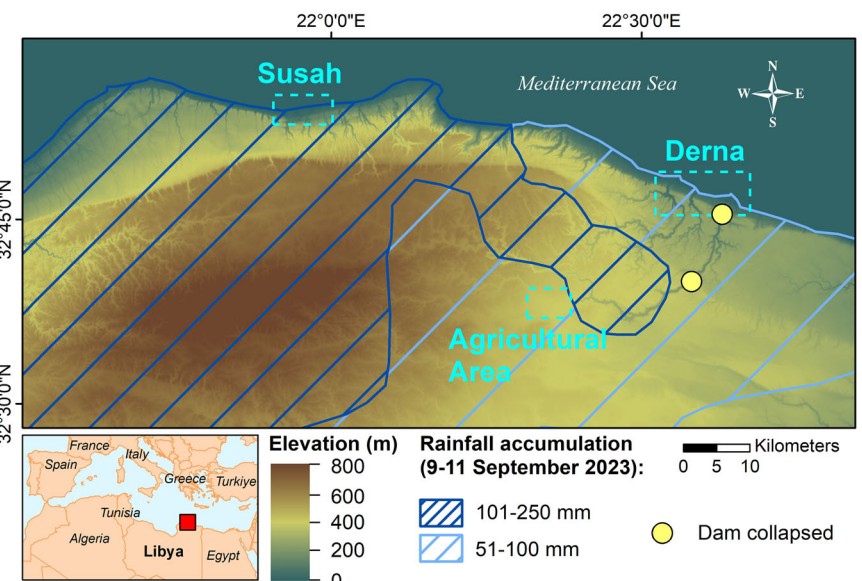

**Fig. 1 | Topography (NASA DEM) and Storm Daniel's rainfall accumulation (9–11 September 2023) within the Region Of Interest (red square).** The rainfall accumulation map was produced by the Emergency Response Coordination Centre (ERCC) (https://erccportal.jrc.ec.europa.eu/ECHO-Products) and derived from NASA Global Precipitation Measurement (GPM)[74]. The three cyan squares represent the cities of Derna, Susah, and an agricultural floodplain, which are further examined in our investigation. Moreover, we indicate the two dams that collapsed (in yellow) along the Wadi-Derna, which triggered the catastrophic flash flood in Derna.

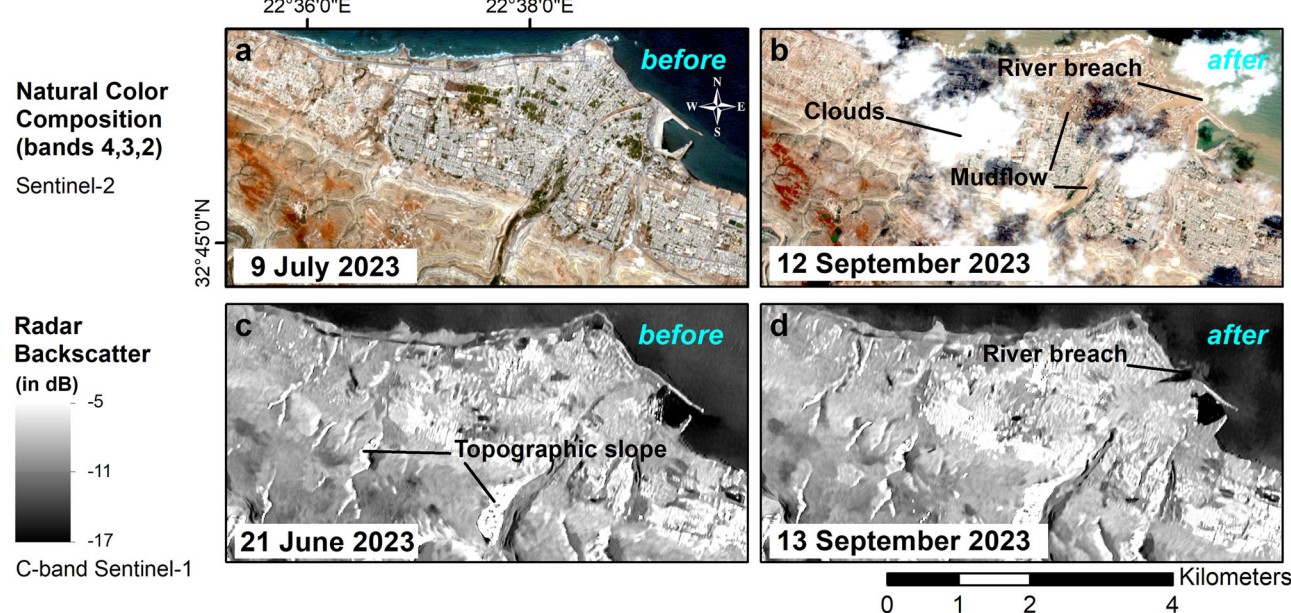

**Fig. 2 | Before and after comparison of flood impact in Derna using visible and SAR amplitude images.** Multispectral (Sentinel-2) and SAR (Sentinel-1A) images comparing the city of Derna before and after the flooding. **a** Visible image of 9 July 2023; months before the flood. **b** Visible image of 12 september 2023; the day after the flood. **c** SAR amplitude image of 21 June 2023; months before the flood. **d** SAR amplitude image of 13 September 2023; two days after the flood. Cloud coverage poses a major challenge for optical imagery, whereas SAR effectively penetrates clouds. However, the post-storm SAR amplitude image does not show major changes compared to the pre-storm image.

soils[11]. Despite the limited urban footprint in the storm landfall area[10], substantial infrastructure damage has been observed, including 5% of the road network being destroyed, 50% becoming inaccessible[10], and the collapse of two flood-control dams upstream of the city of Derna on September 11th[12] (see Fig. 1). The latter triggered a deadly flash flood flowing through the city, located on the river's deltaic outlet (see Fig. 2), destroying a substantial portion of the city buildings, urban infrastructure, and bridges[13], which resulted in the accumulation of 8.8 million tons of debris[14]. For instance, 10% of the houses were destroyed, and 18.5% were damaged[10]. The storm notably affected the city of Derna but also other coastal cities. For example, 28% of the houses in Susah were destroyed[10]. Albayda, Al-Marj, Shahat, Taknis, Battah, Tolmeita, Bersis, Tokra, and Al-Abyar have also witnessed severe damages, causing a total of 5898 casualties, 8000 missing, 44800 people displaced, and 18838 houses damaged or destroyed in the whole coastal area encompassing Benghazi, Jabal Al Akhdar, Al

Marj and Derna[14] (see Figs. 3 and 4). These figures mark Storm Daniel as the deadliest storm in all the African continent since 1900[10].

This eastern coast of Libya is one of the main hubs for energy export to Europe. This catastrophe will cost US$ 1.8 billion in repairing the coastal damages[10]. This figure includes the devastation of crops and harvests across 16209 ha, along with the destruction of vital topsoil for 4.6% of the agricultural area[10]. Additionally, 74363 animals perished, accounting for 3.2% of the region's livestock strategic reserve[10].

Furthermore, this catastrophe is worsened by pre-existing humanitarian needs and restricted road access due to political instability and intermittent civil wars since 2011[15]. These factors impede the reconstruction efforts, potentially leading to regional negative economic growth over the coming years and delaying the recovery of the national economy[10].

As such, characterizing the vulnerability of watersheds and their outlets is essential for effective flood management, avoiding further

humanitarian disasters in this instable area. However, only limited mapping of the impacts of Storm Daniel was explored, focusing mainly on the coastal urban areas and neglecting the inland parts constituting the watersheds. For instance, a few days following this catastrophic landfall, Copernicus Emergency Management Service (CEMS) conducted 'rapid flood' mapping from photo-interpretation for the cities of Derna, Benghazi, Al Marj, Al Bayda, Shahat, Tobruk and Susah. They delineated the flood extent and delivered an assessment emergency mapping[16]. Then, the European Union, United Nations, and World Bank Group collaborated to publish, in January 2024, a comprehensive report on the catastrophe[10], including 'flood intensity mapping' and the flooding extent. In their report, they emphasize the limited availability of ground-truthing data, leading to the use of social media analytics, cell phone data, and night light satellite observations. However, these efforts did not characterize the flood-related soil erosion within the watershed, generating dense turbid flows, which modulate the damages of the catastrophic floods.

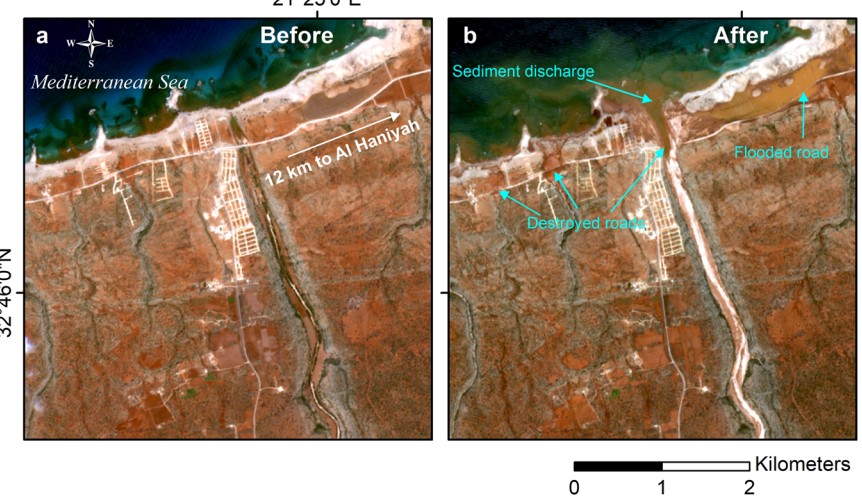

**Fig. 3 | Before and after comparison of flood impact near Al Haniyah using visible images.** Watershed outlet located 12 km West of Al Haniyah and 20 km West of the ROI. The Sentinel-2 images are acquired before Storm Daniel on 2023−09−07 (**a**) and after on 2023−09−15 (**b**). The flash flood occurring during Storm Daniel destroyed and flooded coastal roads in several places.

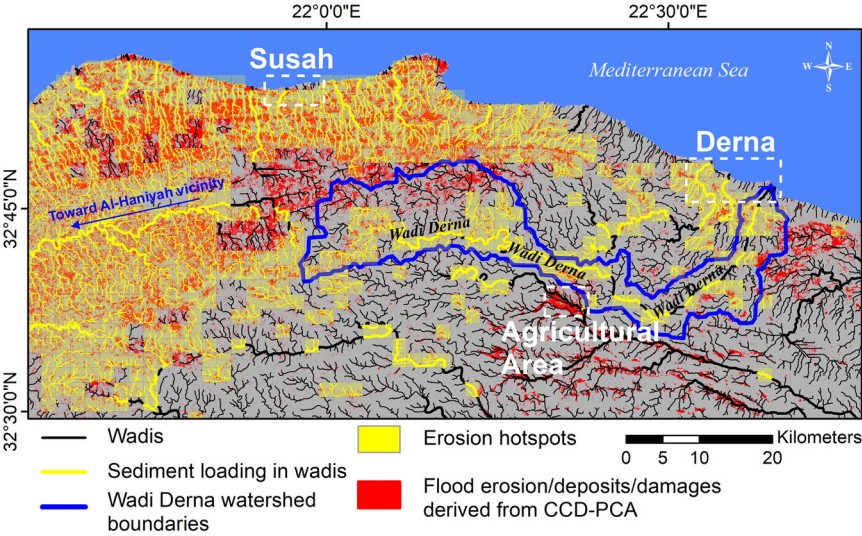

**Fig. 4 | Flood surface change (in red), erosion hotspots (in light yellow), and sediment loading in wadis (in yellow) resulting from Storm Daniel and derived from our CCD−PCA analysis.** Erosion hotspots and sediment loading in wadis contribute to the sediment balance, which increase the likelihood of future mudflow events during rainstorms. Notably, the dense turbid water flow within the Wadi-Derna contributed to the collapses of two dams upstream of Derna city.

To better understand the role of flash flood erosion and further leverage the above-mentioned preliminary investigations, we perform a comprehensive flood surface change mapping at watershed scale levels rather than focusing solely on the small damaged urban areas at the watershed outlets. We achieve this objective by utilizing Synthetic Aperture Radar (SAR) satellite scenes to produce interferometric (InSAR) coherence images and applying the well-established Coherent Change Detection (CCD) method, which measures the temporal variations between InSAR coherence images[17]. The method used herein offers the advantage of assessing structural damages and soil erosion at a level of detail and resolution beyond what is achievable with the optical images used in the preliminary efforts described above.

This CCD approach enables the identification of changes in spatial patterns, such as monitoring flood erosion[18–20], displacement of surface scatterers, and sediment deposition in run-off channels[21], but also to assess different levels of infrastructure damages after a rainstorm[22,23]. These changes can be attributed to both erosion and sediment load processes. Given the inherent interconnection between the two processes, we frequently use the term "flood surface erosion mapping" in our investigation to designate both processes. However, we can delineate the sediment loading areas by conducting a comprehensive classification. This approach relies on erosion hotspots identified through CCD analysis, hydrological networks, precipitation and topography. Moreover, our investigation uses SAR images acquired under dry conditions to map the combined accumulation of (i) flooding surface erosion, (ii) sediment loading in run-off channels, and (iii) infrastructure damages in urban areas.

Using a series of eight SAR Sentinel-1 SLCs, we generate five pre-storm and two co-storm (i.e., overlapping the storm event) coherence pairs. Subsequently, we perform a principal component analysis (PCA) to reduce the dimensionality of this coherence dataset and extract temporal differences. These differences serve as a proxy for mapping flood erosion and assessing infrastructure damages. The complete description is provided in the Methods section at the end of the text, and additional information can be found in the Supplementary Information.

From this CCD–PCA analysis, we map the surface changes over an area of 115 km × 40 km and showcase the cities of Derna, Susah and a cultivated floodplain. Our results and discussion pertain to the following:

- Mapping of flood erosion at watershed scale;
- Mapping of sediment loading in wadis;
- Mapping urban damages in Derna and Susah and classifying their magnitudes;
- Mapping of flash flood sediment imbalance on watershed outlets;
- Discussing the above's implications for other arid, low-lying urban coastal areas.

## Results and discussion

Our findings pertain to using radar InSAR coherence to identify and map the surface and structural changes that are often unmeasurable from optical satellite observations, as detailed in the Methods section. In particular, we measure these changes at different scales, including watersheds and hillslopes (Fig. 4), watershed outlets (Fig. 3), coastal urban areas such as the cities of Derna (Fig. 5) and Susah (Fig. 6), and inland agricultural developments (Fig. 7), as detailed below. Notably, we show the extents and magnitudes of structural damages in the above urban areas, flood surface erosion within the watersheds, and sediment deposits in the wadis caused by the flash flood resulting from Storm Daniel. Finally, we discuss the implications of the above catastrophic processes for other arid areas.

### Flood erosion at the watershed scale

Rainstorms in coastal arid watersheds, such as the one of Wadi-Derna (outlined in blue in Fig. 4), generate run-off water that converges into a network of wadis, discharging toward a common outlet at the coast. These flows are loaded with sediments resulting from upstream soil erosion, which increases the destructive nature of these flash floods. Thus, outlet areas are particularly exposed to the above hydraulic hazards that are often mitigated with a sequence of upstream dams. However, the two dams upstream of Derna have collapsed due to the anomalous hydraulic nature (i.e., flow rate and turbidity) of the floods resulting from Storm Daniel, and lack of maintenance. The extent of the damages triggered by these energetic floods and the associated dam failures are illustrated in Fig. 5. Therefore, assessing the soil erosion and sediment loading in the watersheds is essential to understand the severity of the damages.

Consequently, we apply the CCD–PCA method within the region of interest (ROI) to identify and map the soil erosion areas (see Fig. 4). Notably, the identified erosion patterns show higher occurrences where rainfall accumulation is the highest, i.e., 101–250 mm for the three days from 9–11 September 2023 (see Fig. 1). Moreover, we observe that ~22–26% of the rangelands (21.6% for croplands and 25.8% for shrublands), ~11% of the bare soils and ~11% of urban areas within the ROI are affected by flood erosion, deposits or damages derived from CCD–PCA (see section "Comparison of the flood erosion/deposits with the land cover" in the Supplementary Material for more details). Additionally, these erosion patterns are principally

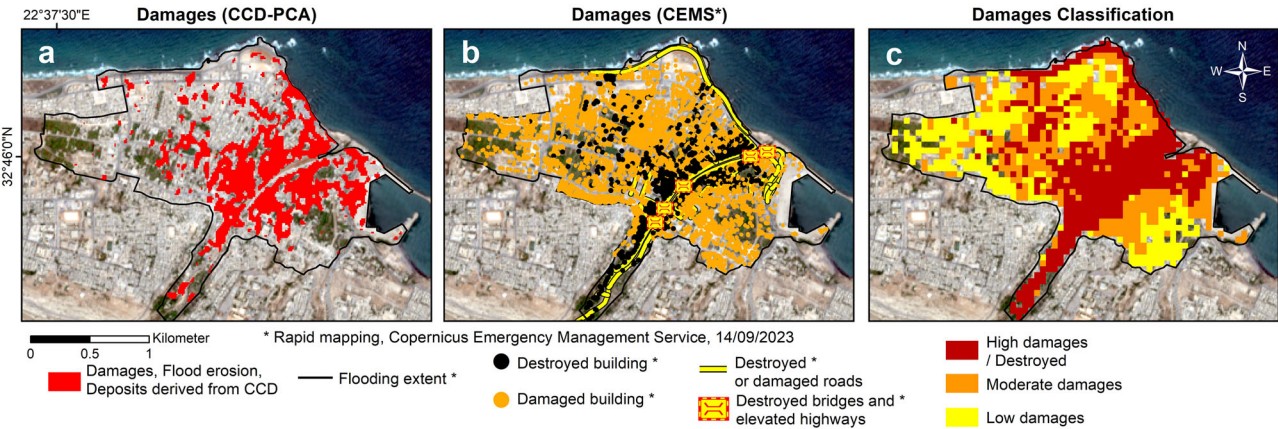

**Fig. 5 | Classification of damage severity (high, moderate and low damages) arising from the flash flood in Derna. a** Damages identified with CCD–PCA. **b** Damages identified by the Copernicus Emergency Management Service[16] (CEMS). **c** Damages classification determined by intersecting our CCD–PCA product with the CEMS dataset within 50 m × 50 m spatial units.

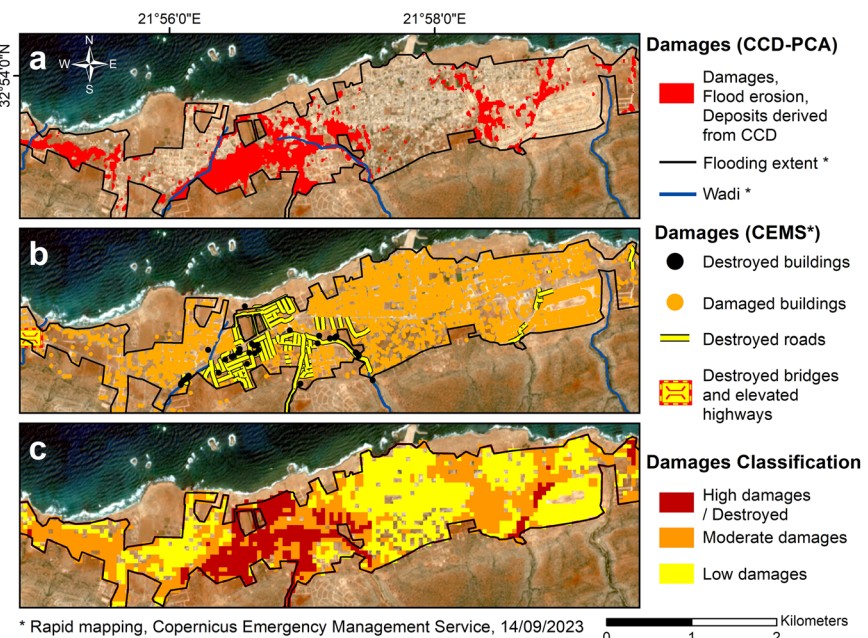

**Fig. 6 | Classification of damage severity (high, moderate and low damages) arising from the flash flood in Susah. a** Damages identified with CCD–PCA. **b** Damages identified by the Copernicus Emergency Management Service[16] (CEMS). **c** Damages classification determined by intersecting our CCD–PCA product with the CEMS dataset within 50 m × 50 m spatial units.

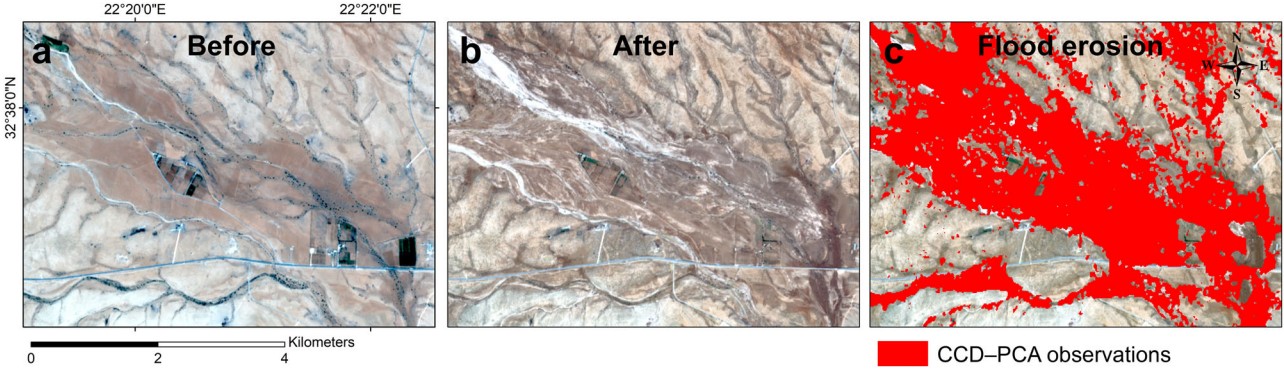

**Fig. 7 | CCD–PCA observations over an agricultural area. a** Pre-storm optical image (RGB composition from Sentinel-2, acquired on 2023-07-09). **b** Post-storm optical image (acquired on 2023-09-12). **c** Flood erosion identified from CCD–PCA (in red) within a cultivated area.

concentrated in areas with elevations ranging from 500 m to 800 m (see Fig. 1) characterized by a slope under 15°.

We find that erosional processes vary notably depending on each of the investigated watersheds (see erosion distribution in Fig. 4 and watershed boundaries in Fig. 8). For instance, in the Wadi-Derna watershed, surface changes mainly occur within the wadis, on moderate-slope (gradient of 4.9° on average) and bare areas, whereas in watersheds located in the hilly northern part of the ROI, surface changes are widespread across the shrublands. Further below, we explore the distribution of erosion hotspots, where the loading of eroded material occurs upstream in wadis.

Furthermore, in arid regions, farmers cultivate areas situated in wadi floodplains, where the topsoil presents favorable conditions for farming. Floodplains are characterized by their rich soil quality, notably marked by a substantial organic matter content, especially within the dammed wadis floodplains[24], as commonly observed in this region of Libya. Moreover, floodplains also present higher soil moisture and finer soil particles[24], and they often provide access to subsurface water

resources for irrigation. This groundwater arises from focus recharge during rainstorms, replenishing subsurface freshwater lenses beneath wadis[25].

Flash floods, however, outwash wadis and their floodplains, eroding these fertile soils[24]. Notably, Fig. 7 showcases an inland cultivated floodplain before and after the flood caused by Storm Daniel, revealing the changes from cloud-free optical satellite images and CCD–PCA analysis. Although visual examination of the optical images delineates the extent of flood surface erosion affecting a substantial area spanning 7 km × 2 km, our CCD–PCA observations offer "automated" surface change detection with enhanced sensitivity, providing finer details. From this detection, we can assess the impact of the water flow over vulnerable farms.

These flood erosions restrict the long-term sustainability of farming developments, thereby impacting food security for the local population. Thus, it is imperative to implement comprehensive flood management strategies to mitigate flood hazards posed to crops facing changing environmental conditions.

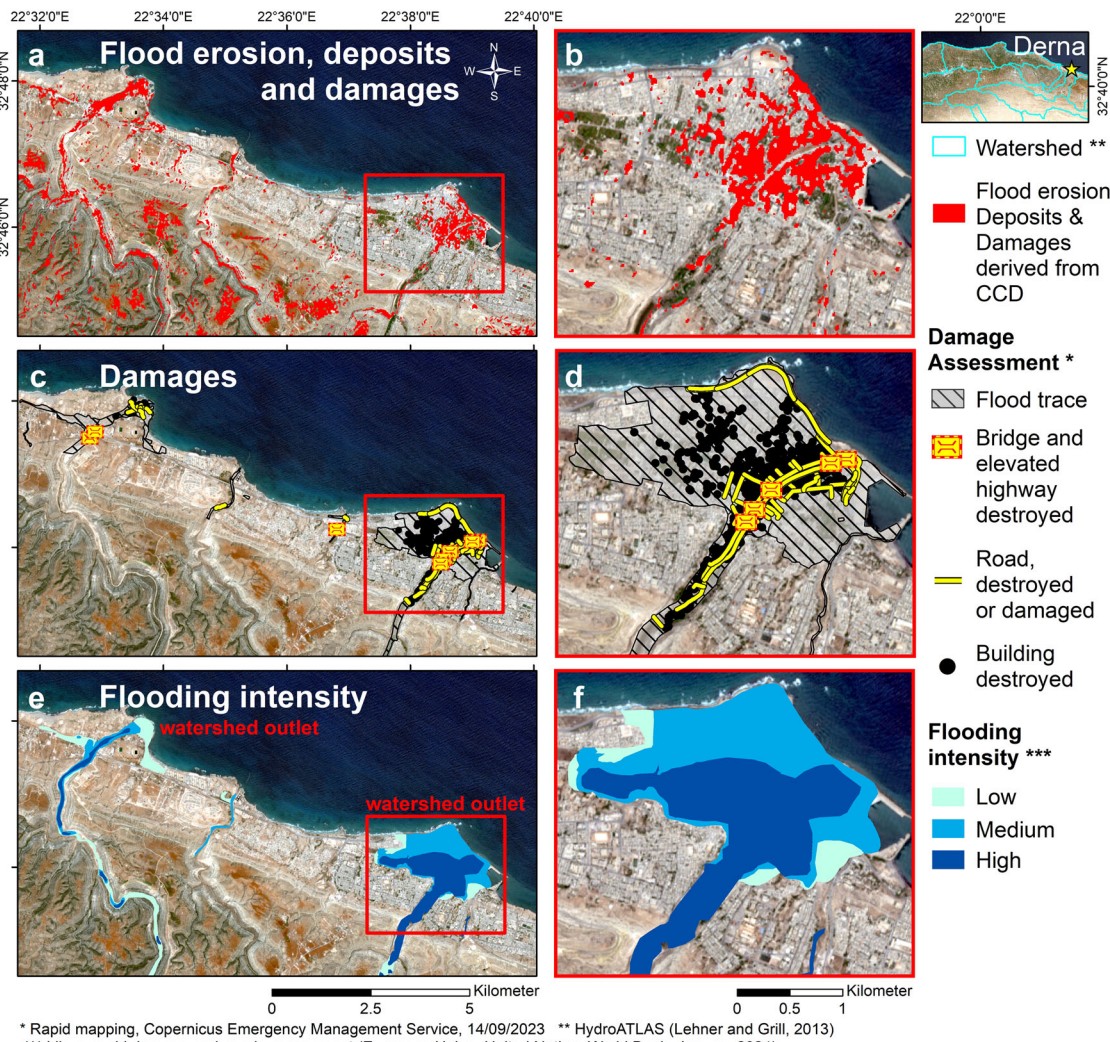

**Fig. 8 | CCD–PCA observations in Derna. a** Identification of flood erosion sediment loading and building damages (in red) through CCD–PCA over two watershed outlets in Derna. **b** same as (**a**) but over Derna city center. **c** Identification of flood traces and infrastructure damages through photo-interpretation by the Copernicus Emergency Management Service[16]. **d** same as **c** but over Derna city center. **e** Modeled flooding intensity derived from a 12-m DSM (European Union, et al. [10]). **f** same as (**e**) but over Derna city center. Our CCD–PCA results corroborate with the damage assessment and flooding intensity model. The Sentinel-2 pseudo-image underlying the maps was acquired on 2023-07-09.

## Sediment loading in wadis

In semi-arid regions, precipitation, in the form of scarce and heavy rainfalls, leads to rapid surface run-off within slopes, hillslope sediment fluxes, erosion and sediment loading in the wadis, valleys, or low-lying areas[26–28]. In this context, lithology, erodibility, slope gradient[29,30], and land use are the main factors characterizing surface run-off, erosion and sediment load[31].

We perform a first-order classification of this run-off erosion and sediment balance dynamics on steep gradients above 3° and hillslopes, where loose sediments are washed away and subsequently deposited below in wadis, increasing the likelihood of future mudflow events during rainstorms. Notably, Fig. 4 indicates erosion hotspots and sediment loading after Storm Daniel, in the North and West parts of the ROI, characterized by hillslopes and shrublands (see Supplementary Fig. 1). Moreover, similar observations are made within Wadi-Derna. The watershed of Wadi-Derna features an average terrain gradient of 4.9° (±5.5), characterized by exposed bare soil that facilitates overland flow throughout the area towards the wadi. This dense turbid water flow within the wadi further contributed to the collapses of two dams upstream of Derna city.

Moreover, we identify a substantial sediment loading zone within a wadi flowing towards the vicinity of Al-Haniyah, situated outside the ROI. While this outlet was not initially the focus of our study, we have chosen to illustrate it in Fig. 3. It illustrates the conditions before and after the storm, showing destroyed and flooded segments of the sole coastal road near the wadi's mouth.

## Urban damages in Derna and Susah

Overall, our CCD–PCA mapping reveals that 18% of Susah's surface area exhibits damage and flood surface changes. This percentage is determined by calculating the ratio of the CCD–PCA flood surface change to the total surface area within the city limits. Similarly, in Derna, two watershed outlets present notable damages, erosion and deposits, particularly within the wadis and deltas. Figure 8 shows the extent of damages within the city and its surroundings, derived from CCD–PCA, covering 9% of its surface area and 27% of Derna's city center. These observations are validated with two distinct sources, showing a true-detection of 61–66% and 42–65%, a false-detection of 7–29% and 18–34%, and an omission error of 6–32% and 2–40%, for Derna's city center and Susah, respectively (see "Validation of

CCD–PCA" in the Methods section for more details). These figures exhibit comparable ranges of value for both cities, reflecting a constant degree of validation of the CCD–PCA method.

On the other hand, these figures underscore CCD–PCA's complementarity nature when compared to the CEMS dataset presenting subjective damage categories, i.e., CEMS classified damages as "destroyed" and "damaged" without providing the metrics used for such visual interpretation. Given the implication of the latter statement, we find it necessary to classify the damage severity for the two cities by intersecting the CEMS damage assessment with our CCD–PCA surface changes observation. Our comprehensive classification now effectively distinguishes between areas of high, moderate, and low damage. The corresponding damage severity maps offer the large-scale snapshot capability of SAR remote-sensing from which the damage magnitude is unclassifiable, with the qualitative capability of CEMS' photo-interpretation and ground-truthing. The assumptions and criteria to perform this classification are detailed in the "Classifying damages in Derna and Susah" section in Method.

We perform this classification within the flooded extent of Derna's city center and observe that the damage severity for high, moderate, and low categories accounts for 36.5%, 29%, and 27% of the surface area, respectively, with 7.5% remaining undamaged (see Fig. 5). Moreover, the same classification within the urban area of Susah, as outlined in Fig. 6, reveals that 14% of the surface area exhibits high damages, 34% displays moderate damages, 39% shows low damages, and 13% remains undamaged.

This enhanced classification is an essential tool that could facilitate on-site damage assessment, coordination of humanitarian assistance and reconstruction efforts and aid decision-making for flood management.

## Flash flood sediment imbalance at the watershed outlets

Extremes rainstorms, such as Storm Daniel, contribute to an intermittent and limited sediment supply from wadis to the eastern Mediterranean basin's sandy shorelines. This limited sediment input, coupled with urbanization, waterways, bridges, and upstream dams, disrupts the wadi flow[28,32], which results in shoreline retreat, beach losses and coastal flooding during storms and surges[33,34]. Moreover, marine currents, tide force, and wave energy[35] are additional factors driving this coastline dynamic.

Other drivers contribute to the shoreline retreat, such as unregulated and excessive sand mining[33], sea-level rise[36], relative sea-level rise arising from coastal ground subsidence[37–42], and weakening of the protective coastal barrier. The latter comprises sandbars (i.e., submerged sand ridges), extensive beach width, and dunes, constituting the primary defense against storm surges[43,44]. However, the combined forces of wind, waves, and rip currents can considerably modify the beach's geomorphology, locally compromising this protective barrier.

Furthermore, the increasing aridity of the region presents an ambivalent impact on sediment supply in wadis: it not only diminishes water discharge and sediment load in wadis but also contributes to desertification, reducing vegetation that retains sediments, thereby amplifying sediment erosion and loading wadis[45].

Globally, the northeastern coast of Libya indicates no substantial coastal erosion for 1984–2016, with some exceptions, including a 3.2 m yr$^{-1}$ (±0.6) regression in Al Haniyah, 1.2 m yr$^{-1}$ (±0.3) accretion at the mouth of Wadi-Derna, and 1 m yr$^{-1}$ (±0.3) regression in an isolated beach in Susah[34]. The latter regression in Susah is consistent with previous findings in Westley and Andreou[33] and is discussed further below in this section.

Prior to Storm Daniel, Susah suffered locally from coastal erosion, damaging the Hellenistic and Roman archaeological sites of Apollonia, its historic name[33,46]. For instance, Fig. 9 shows that the shoreline had retreated by ~200 m since 1974[33], or ~50 m according to Luijendijk, et al.[34], precisely at the mouths of two wadis in Susah, revealing a

disappearance of the protective barrier and/or a lack of accretion. This regression is likely associated with a decrease in the frequency and/or magnitude of rainstorms, changes in land use, and, therefore, a decrease in sediment discharge from these wadis into the sea. However, further investigations are needed to validate this hypothesis of decreased sediment supply impacting the coastline. Moreover, a better understanding of the complex interplay between local environmental factors characterizing arid regions, the influence of regional weather patterns, and the eastward shift of the medicanes, typically located in the western and central Mediterranean Sea[1,2], is essential.

Our CCD–PCA analysis in Susah reveals infrastructure damages within the city and flood erosion at the wadis' mouths, as illustrated in Fig. 9. The resulting flooded or swept-away roads led to Susah's isolation on September 13th, 2023[47]. In particular, the largest damage area is located at the confluence of two wadis where sediment discharge, at their mouths, potentially accreted and replenished the beach abovementioned and protective barriers. Notably, we observe the apparent reactivation of a wadi within Susah's delta, which had been obstructed since before 1974 (see Fig. 9). This reactivation partially confirms the hypothesis of localized shoreline retreat that started after a flow obstruction resulting from infrequent rainstorms and slow sediment loading within the wadis.

Episodic weather patterns intermittently obstruct wadi's flow for extended periods, spanning several decades, for example. Such prolonged intervals affect public perception, leading the population to underestimate the risk[48] and construct infrastructures and buildings in these vulnerable areas.

## Implications for other arid coastal areas

Desert floods are widely observed in other arid areas, such as the desert of the Arabian Peninsula, Sinai (Egypt), Oman, Jordan, Israel, and Qatar, among others[21,49–54]. However, quantitative damage assessments of flash flood aftermath are uncommon. When such investigations are conducted, they are primarily conducted through optical photo-interpretation. This technique is typically applied in localized areas for humanitarian purposes, aiming to assess damages in urban communities to support rescue or rehabilitation efforts. However, they often fail to assess the drivers accentuating some of these events. As demonstrated in our CCD–PCA analysis, mapping damages and flood erosion at a larger scale offers various advantages, such as a more comprehensive understanding of sediment loading in wadis, identifying erosion and accretion processes, and identifying damaged crops in floodplains. As such, mapping the extent of the post-storm erosion is crucial to understanding floods and sediment transport dynamics to update existing urban plans and coastal management strategies to be more resilient to future hydroclimatic extremes.

Gravity dams, like those that collapsed in Derna, play an essential role in flood management in arid regions. These structures are designed to withstand rapid changes in hydrostatic pressure caused by changes in environmental load conditions, such as successive droughts and flash floods that can result in deformation and cracking[55]. Their maintenance is critical to ensure their resilience and prevent potential failures. Consequently, there is a need for fragility assessments, particularly for the dams situated in catchments showing extended erosion hotspots and sediment loading that can exacerbate the changes in hydraulic pressure caused by dense turbid water, as outlined in our investigation.

Furthermore, the primary importance of accurately mapping the flood erosion on the watershed scale is assessing the local drivers amplifying the devastating nature of flash floods under increased hydroclimatic fluctuations in the eastern Mediterranean basin and the Arabian Peninsula[56]. This, in turn, helps mitigate adverse effects on human lives, infrastructure, water management, agriculture, and other vital socio-economic activities in these harsh desert environments. For instance, the urbanized watershed outlets of other coastal areas such as

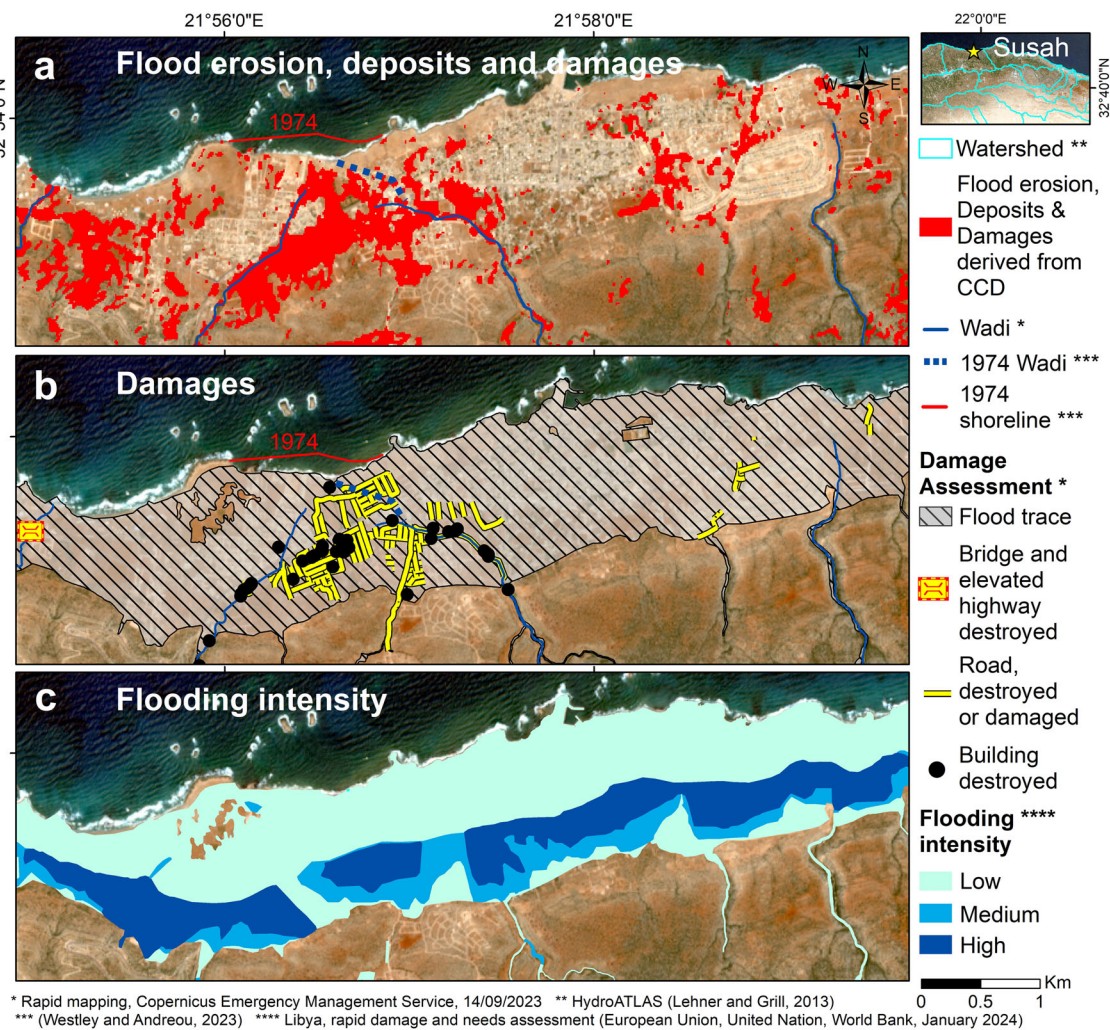

**Fig. 9 | CCD–PCA observations in Susah. a** Identification of flood erosion sediment loading and building damages (in red) through CCD–PCA in Susah. **b** Identification of flood traces and infrastructure damages through photo-interpretation by the Copernicus Emergency Management Service[16]. **c** Modeled flooding intensity derived from a 12-m DSM (European Union, et al.[10]). Our CCD–PCA results corroborate with the damage assessment and flooding intensity model. Moreover, the 1974–shoreline has not been accreted since the early 1970s due to the deactivation of a channel (blue dash line) (Westley and Andreou[33]). This specific coastal area was heavily affected by the water flow, potentially leading to the reactivation of the wadi mouth. The Sentinel-2 pseudo-image underlying the maps was acquired on 2023–07–09.

Medjerdah in Tunisia, Tripoli in Libya, Alexandria in Egypt and south-western of the Arabian Peninsula face increased flood events with several recorded damages[32]. Using the CCD method in these areas following the next rainstorms will be crucial for understanding the extent and magnitude of the watersheds' physical attributes, particularly in the context of increased droughts and floods. Evaluating the linkage between these physical characteristics and the resulting flood damages solely through optical photo-interpretation poses multiple challenges. Therefore, it is imperative to expand SAR coverage and CCD implementation across vulnerable coastal arid areas for the upcoming decades to comprehend the convoluted relation between the occurrence of these hydroclimatic extremes and their impacts in these poorly characterized yet densely populated areas. It is also essential to understand how these extremes affect coastal morphodynamics.

Finally, by implementing proper infrastructure and understanding sediment transport processes within watersheds, these hydraulic extremes can be leveraged to harvest stormwater, relieving local water scarcity. Using this stormwater can facilitate the creation of green spaces[57], minimizing soil erosion and improving the drainage of future stormwater, among other benefits such as urban agriculture[58] and reduction of heat islands in cities[59]. Our findings suggest that redesigning a substantial part of the coastal cities in arid areas is necessary to improve their resilience to extreme events. This conclusion is evidenced by the severity of observed damages across Derna city center and Susah, typical expanding deltaic cities in the eastern Mediterranean basin, revealing that 66% and 48% of their respective surface area have experienced moderate-to-high damages. These figures underscore the need for urban planners and policymakers to rethink the urban fabric of these coastal cities, where hydroclimatic extremes, though rare and sparse, are often underestimated[48,60]. As such, we call for the urgent need to undertake strategic land-use transformations in several of these areas, notably by creating green spaces to control soil erosion[61]. Implementing such policies under the current socio-economic instabilities and the lack of public awareness of hydroclimatic risks[48] remains a challenge.

## Methods
In this section, we first expose the gained value in using SAR radar rather than optical images to identify surface changes after rainstorms in arid and semi-arid areas. Then, we introduce the CCD–PCA method for producing the flood erosion map. Additionally, we leverage the resulting flood erosion map to evaluate the erosion hotspots, an

important step to estimate sediment loading spatially in wadis. This latter assessment is essential as sediment load intensifies the turbidity of flash floods and influences the likelihood of mudflow occurrences in watershed outlets during future rainstorms. Moreover, we classify damage severity in the cities of Derna and Susah into three levels using the CCD–PCA method and ground-truthing dataset from CEMS. Finally, we elaborate on the validation and limitations of the CCD–PCA method.

## Gained value in using radar observations

Figure 2 shows the before/after map of the flooding in Derna, captured by Sentinel-2 multispectral and Sentinel-1A SAR satellites for the summer and fall of 2023. Storm Daniel's extensive cloud coverage obstructs satellite optical observations, yet some parts of the city reveal traces of mudflow. In contrast, SAR's ability to penetrate the clouds enables the assessment of the entire region. For instance, on the SAR amplitude image, we identify the river breach in Derna's delta resulting from the characteristic specular reflection of the radar wave on the water. However, it is challenging to visually assess the building damages, flood surface erosion, sediment deposits, and flooded extent from SAR amplitude images only due to the relatively lower spatial resolution compared to the optical ones.

Moreover, saturated soils exhibit higher dielectric properties, producing high radar amplitudes. Thus, wet flooded extents can be identified using the Non-Coherence Change Detection (NCCD) technique[23,62]. NCCD is performed by comparing SAR backscatter amplitudes between acquisitions to identify specular reflections on flooded areas and increases in dielectric on wet soils. To this end, we compare the backscattering amplitudes between the "dry" acquisition of 2023–06–21 and the "wet" acquisition of 2023–09–13 (see Fig. 2); however, we do not observe significant dielectric changes within the city. Notably, we calculate a negligible amplitude increase of 0.5 dB after the flash flood, with a median value of −7.1 dB for 2023-06-21 and −7.6 dB for 2023-09-13, over the flooded city. Therefore, using NCCD is not relevant in our investigations.

Furthermore, unlike SAR amplitude images, Single Look Complex radar (SLC) data products comprise both amplitude and phase information. The phase is crucial when assessing changes in surface scattering between acquisitions. Notably, the surface damages and erosional processes caused by a flash flood alter the phase spatial distribution of the scattered radar returns between pre and post-storm acquisitions over the impacted area. Moreover, the temporal variability of the radar phase is described in (1), (2), and (3); these equations encompass the contribution of the phase difference in the interferometric SAR (InSAR) coherence that uses SLC SAR images.

Therefore, the NCCD above-mentioned example over Derna and radar scattering principle underscore the rationale for using SLC SAR images rather than SAR amplitude images to observe flood aftermaths in arid regions.

## CCD–PCA method

The initial step of the CCD–PCA approach involves producing a series of InSAR coherence images. These images are derived from SLC SAR scenes that contain amplitude and phase information for each pixel, which varies with soil dielectric and surface scattering. Therefore, by comparing the amplitude and phase of two distinct co-registered SLC SAR scenes representing the same footprint at different dates, we can assess the surface changes. The spatiotemporal variability of the amplitude and phase is estimated by calculating the cross-correlation, or InSAR coherence $\gamma$, between two SLC scenes, as expressed in (1) and (3)[63].

$$\gamma = \frac{\left|\langle c_1 c_2^* \rangle\right|}{\left(\langle c_1 c_1^* \rangle \langle c_2 c_2^* \rangle\right)^{1/2}}, 0 \leq \gamma \leq 1 \qquad (1)$$

$$with \ c = A e^{i\phi} \qquad (2)$$

$$\gamma = \frac{\left\langle A_1 A_2 e^{i(\phi_2 - \phi_1)} \right\rangle}{\left(\left\langle |A_1|^2 \right\rangle \left\langle |A_2|^2 \right\rangle\right)^{1/2}}, 0 \leq \gamma \leq 1 \qquad (3)$$

Where $\gamma$ is in the InSAR coherence or complex correlation coefficient, c is an SLC SAR image, $c^*$ is the complex conjugation of c, $c_1$ and $c_2$ are the co-registered complex SAR reference and secondary images, respectively, $A_1$ and $A_2$ are the amplitudes, and $\phi_1$ and $\phi_2$ are the phases of these complex SAR images. Moreover, the angle brackets represent the kernel window where coherence is estimated through spatial averaging using an adaptive nonlocal–InSAR (ANL–InSAR) filter[64].

When computing InSAR coherence between pre-storm and post-storm SAR images acquired under dry conditions, we capture the changes in surface scattering relative to flood surface erosion (see acquisitions of the SAR images in Supplementary Table 2). However, it is essential to acknowledge that soil dielectric is assumed to remain constant between the two dry acquisitions; thus the dielectric changes contribution is negligible in the coherence. Moreover, coherence pairs should be cautiously selected, considering temporal and perpendicular baselines, to minimize spatial decorrelation arising from topography and multi-pass satellite acquisition geometry. Although some residuals could remain, the CCD approach mitigates them.

Figure 10 presents a diagram illustrating the CCD method. The selection of InSAR pairs used in this CCD method is an essential part of the approach; they are listed in Supplementary Table 4. First, it helps to reduce spatial decorrelation by selecting pairs showing small perpendicular baselines[65] (e.g., <100 m). Second, our pair selection excludes potential soil moisture signals in the InSAR coherence by selecting only acquisitions under dry soil conditions. Particularly, we produce seven coherence pairs from dry acquisitions with varying perpendicular and temporal baselines ranging from 23 m to 68 m and 12–72 days, respectively. Five of these pairs correspond to pre-storm coherence with temporal baselines of 12, 24, 36, and 48 days. These five pairs serve as 'reference coherences' or 'background coherences' for isolating the magnitude of coherence attributed solely to vegetation and surface roughness. The remaining two pairs overlap the rainstorm event and contain the "change" information that we aim to identify in this study.

The CCD process could be performed by calculating the coherence difference between two pairs of similar perpendicular and temporal baselines, subtracting $\gamma_{post-storm}$ from $\gamma_{pre-storm}$[23]. Alternatively, an averaged version option would be subtracting the average of several $\gamma_{post-storm}$ pairs from the average of several $\gamma_{pre-storm}$ pairs. Instead, we opt for a different approach, using the PCA to synthesize the information from the seven pairs. PCA reduces a dataset's redundancy and linear dimensionality[66]. From this statistical analysis, we extract two significant principal components (PCs) representing linear combinations of the original dataset. However, due to the non-linear nature of coherence decay over an extended period, we limit our analysis to coherence pairs with a temporal baseline of less than 84 days. The choice of this maximum temporal baseline would differ from one climatic region to another. Hence, to reproduce the PCA data reduction process in regions other than eastern Libya and identify a suitable maximum temporal baseline, we propose an approach detailed in the section "Requirement for effective PCA in the CCD", in Supplementary Information.

In our investigation, the first PC represents 92% of the common information within the dataset, and the second PC accounts for 4.5% of the variability that is not captured by the first PC[67,68]. The first PC can be interpreted as an averaged and standardized pre-storm coherence, while the second PC can be interpreted as the flood erosion signature

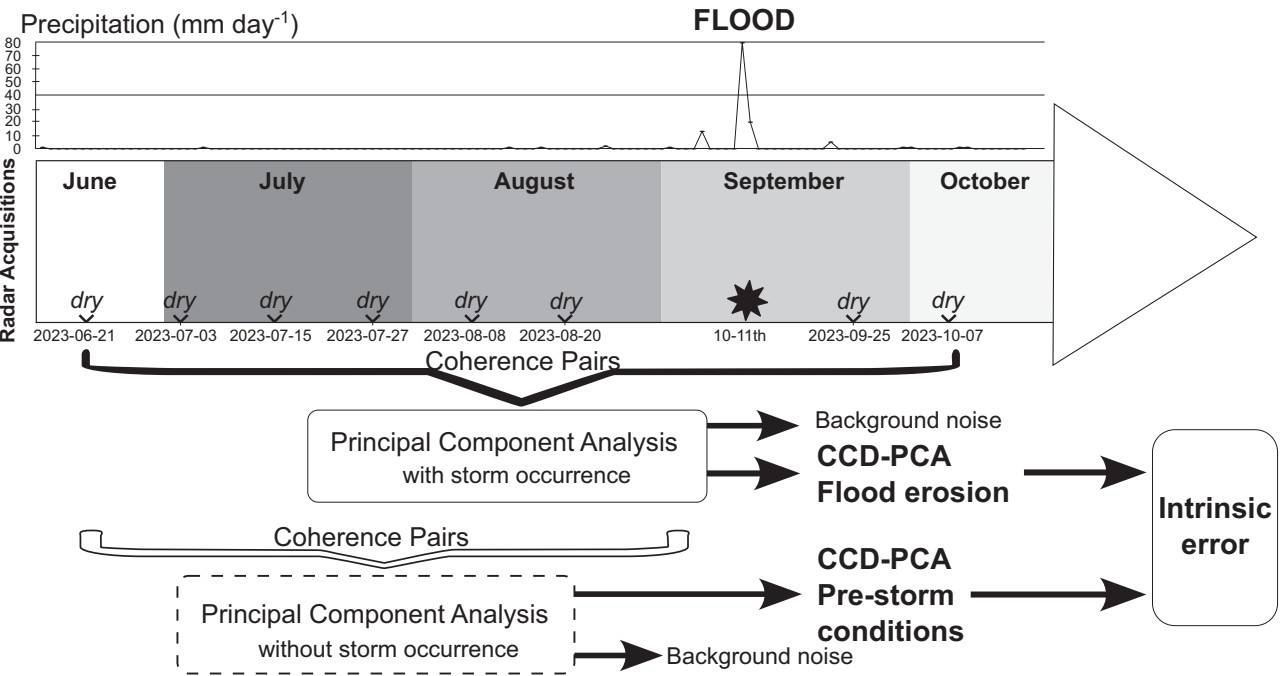

**Fig. 10 | CCD–PCA methodology.** Coherent change detection (CCD) is associated with the principal component analysis (PCA) method to identify the flood erosion occurring during and after Storm Daniel (black star). The timeline displays the dates of the eight SAR acquisitions used to produce the coherence pairs. The coherence pairs are listed in Supplementary Table 4.

(see Figs. 1, 8 and 9). On this second PC, we apply a threshold defined as [mean + δ; maximum coherence value] to facilitate the visual interpretation of our CCD–PCA results and create a shapefile illustrating the post-storm changes.

Furthermore, to test the PCA's sensitivity, its intrinsic error, in identifying any potential erosion preceding the rainstorm and other residual errors, we conduct a similar PCA by excluding the two $\gamma_{post-storm}$ acquisitions above-mentioned (see coherence pairs in Supplementary Table 4). In this scenario, the common information within the dataset constitutes 96% of the information, while the second PC accounts for 1.5% only, characterizing pre-storm erosion. Then, by intersecting this CCD–PCA pre-storm erosion with the CCD–PCA post-storm erosion, we can isolate the false-positive identifications (commission errors) in the CCD–PCA post-storm erosion. This intrinsic error in the CCD–PCA method accounts for a 13% occurrence probability (refer to "Identification of the intrinsic CCD–PCA error" in Supplementary Information for details). This bias identification improves the reliability of our flood erosion mapping process.

Additionally, we use ground-truthing for validation over urban areas, as developed in the "Validation of CCD–PCA" section further below.

### Classifying damages in Derna and Susah

This damage classification is based on the CCD–PCA validation with the CEMS map, detailed in "Validation of CCD–PCA/ Validation using the CEMS damage assessment map" from the Methods section, where both datasets are intersected within a grid, where each grid cell measures 50 m × 50 m. The CEMS dataset encompasses the two following damage scenari:

- Case 1: With damaged or destroyed buildings, destroyed bridges, damaged and destroyed roads
- Case 2: With destroyed buildings, destroyed bridges, damaged and destroyed roads

To conduct this classification, we first assume perfect exactitude in damage detection for both datasets while acknowledging the possibility of damage omission. Moreover, we consider that the CCD–PCA exclusively detects substantial damages or destruction but cannot identify low-magnitude damages. Based on these assumptions, we classify the damages as high, moderate and low according to the following criteria (see Supplementary Table 8):

- High damages: when the damages observed from the CCD–PCA identification are also identified in the CEMS map (Case 1 and Case 2); also when Case 2 damages are not identified (omission) by CCD–PCA:

$$[Case1(True\ detection\ of\ occurrence) \cap Case2$$
$$(True\ detection\ of\ occurrence)] \cup Case2(omission)$$

- Moderate damages: when only CCD–PCA identifies a surface change (false-detection):

$$Case1(False\ detection) \cup Case2(False\ detection)$$

- Low damages: when Case 1 damages are omitted by CCD–PCA; however, we remove from this selection the "High damages" instances of Case 2 damages omitted by CCD–PCA:

$$Case1(omission) - Case2(omission)$$

### Classifying sediment loading

Our assessment of sediment loading in wadis relies on the identification of areas showing high erosion risk or erosion hotspots[69]. To identify these hotspots within the watersheds, we implement a simplified version of the Revised Universal Soil Loss Equation (RUSLE) theoretical method[30]. The RUSLE method involves rainwater erosivity,

soil erodibility, land surface cover, slope gradient and length, and soil erosion management index. Conversely, our CCD–PCA flood surface change detection already integrates the identification of erosion patterns. Thus, we subsequently intersect these CCD–PCA results with 3-day rainfall accumulation and slope gradient variables in order to derive the erosion hotspots (see Supplementary Table 7).

The first variable considered herein is the 3-day rainfall accumulation during Storm Daniel; we chose a conservative estimation threshold of 50 mm, which enables the covering of the whole ROI (see Fig. 1). The second variable corresponds to the post-storm flood erosion, which is a metric derived from our CCD–PCA method. We define erosion as significant when more than 5% of the surface area within any 2 km × 2 km bin shows a CCD–PCA signal (i.e., 0.2 km²). The third variable under consideration is the average topographic slope gradient within a 2 km × 2 km bin; we use the threshold angle of 3°, a common gradient threshold employed in arid and semi-arid regions to characterize erosion[70,71]. Assigning weights to these above-mentioned variables is challenging; therefore, multivariate statistical techniques such as factor analysis (e.g., PCA) can be performed to determine weighting factors[72]. Instead, in our investigation, we apply a first-order binary classification based on thresholding to identify erosion hotspots arising from Storm Daniel. The corresponding classification criteria are addressed in the Supplementary Information's "Sediment loading assessment" section.

Lastly, after characterizing the erosion hotspots, we assess the areas of sediment loading by intersecting the identified hotspots with the locations of wadis.

The intermediary mapping is displayed in Supplementary Information's "Sediment loading assessment" section. Moreover, the methodology used to derive the wadi networks from the NASA DEM is presented in the Supplementary Information's "Identification of wadis" section.

Furthermore, to leverage this erosion hotspots assessment, we recommend applying the same CCD–PCA method but in the ascending direction (opposite to the descending direction used in this study). This second iteration would allow erosion mapping in the radar-shadowed areas and contribute to a more robust assessment of the erosion dynamics within the slopes.

## Validation of CCD–PCA
We validate our CCD–PCA observations over the cities of Derna and Susah through verification with two distinct sources. The first source was published in the European Union, et al.[10] report and represents the flooding intensity map generated from flow modeling derived from a 12–meter resolution Digital Surface Model (DSM). Although this dataset does not consider the density of habitations and the stormwater drain infrastructures, the authors used this model as a direct proxy for assessing the magnitude of damage. Consequently, they determined that 10% and 28% of the housing in Derna and Susah, respectively, were destroyed. These findings align in magnitude with our assessment of surface changes in Derna and Susah, revealing that 9% and 18% of their respective surface areas were impacted (see "Urban damages in Derna and Susah" section for details).

The second source was produced by Copernicus Emergency Management Service[16] (CEMS) and represents the damage assessment evaluated through photo-interpretation, as depicted in Fig. 8. This photo-interpretation reveals distinct categories, including 'flood traces' extents, 'buildings destroyed', 'buildings damaged', 'bridges and elevated highways destroyed' and 'roads destroyed or damaged'—the term 'photo-interpretation' is mentioned in the product metadata. The two categories, 'buildings destroyed' and 'buildings damaged', are subjective and, therefore, influence our validation assessment.

Our CCD–PCA product, in its current state, identifies surface changes without providing precise information about the specific nature of these changes. However, we assume that the CCD detections

within urban areas are primarily attributed to infrastructure damages rather than surface erosion and sediment deposits.

**Validation using the flooding intensity map.** In Derna, the high-intensity flow model corresponds closely to the areas of damage observed on the CEMS damage assessment map and CCD–PCA map (see Fig. 8). In particular, both the CCD–PCA and the high-intensity flooding model accurately delineate the flood extent within the watershed outlets of Derna.

In Susah, neither the damage assessment map nor the CCD–PCA map aligns well with the flood intensity map (see Fig. 9). This shows the limit in comparing datasets of different natures.

**Validation using the CEMS damage assessment map.** In contrast to the previous validation source, the nature and spatial resolution of the CEMS damage assessment map facilitate quantitative validation of the CCD–PCA results. To achieve this for Derna city center and Susah, we intersect the CCD–PCA results with the CEMS dataset within a grid, where each bin measures 50 m × 50 m.

Given the inherent differences between the two maps (i.e., the CCD–PCA is derived from radar scattering while the CEMS map is generated from photo-interpretation), we divide the CEMS dataset into two distinct maps: Case 1 map includes damaged and destroyed buildings, as well as damaged/destroyed bridges and roads, while Case 2 map comprises solely destroyed buildings, with damaged/destroyed bridges and roads.

Within the flooding extent[16] of the city center of Derna, we determine that the CCD–PCA map corresponds to the CEMS-Case 1 map for 61% of the area (true-detection), with a false-detection of 7% and an omission error of 32% (see Supplementary Fig. 2, presented in the Supplementary Material under the "Validation of CCD–PCA in Derna and Susah" section). With the CEMS-Case 2 map, we obtain a true-detection rate of 65.5%, a false-detection rate of 29% and an omission error of 5.5%. The high false-detection rate of 29% in Case 2 and low false-detection of 7% in Case 1 confirm that the CCD–PCA method identifies the CEMS's "destroyed buildings" but is also somehow sensitive to CEMS's "damaged buildings". However, the high omission error of 32% in Case 1 suggests that identifying CEMS's "damaged buildings" remains challenging and is subject to the subjective interpretation of the term "damaged" within the CEMS dataset. Moreover, we suggest that the omission and false-detection errors arise from three possible causes: inaccuracies in the CEMS map, extensive sediment deposits in the streets, or intrinsic error in the CCD–PCA analysis (see "CCD–PCA method" section above).

We perform the same validation over the flood extent of Susah and obtain for Case 1 a true-detection rate of 42%, a false-detection rate of 18%, and an omission error of 40%. For Case 2, we determine a true-detection rate of 64.5%, a false-detection rate of 34%, and an omission error of 1.5% (see Supplementary Fig. 3). These percentages observed in Susah are comparable to those for Derna, suggesting similar sensitivity of radar scattering to infrastructure damage magnitude observations in both cities.

We attribute this CCD–PCA sensitivity to damage magnitude to the difference in phase spatial variability. Specifically, "destroyed buildings" are associated with substantial phase spatial variability, while "damaged buildings" exhibit a more subtle phase spatial variability.

We find that our CCD–PCA damage map aligns well with the CEMS dataset for both cities, achieving a true-detection rate of up to ~65% of the surface area. Furthermore, we suggest that the distinct multitemporal InSAR coherence sensitivity to 'damaged' and 'destroyed' infrastructures enables the derivation of enhanced damage severity maps from CCD–PCA and CEMS datasets. This is achieved in the "Urban damages in Derna and Susah" section (see Figs. 5 and 6).

## Limitations

We identify two sets of limitations restricting our flood surface change mapping. The first set affects the background coherence in the CCD–PCA Methods section. These limitations arise from the influence of the topography in the CCD, introducing spatial decorrelation related to the multi-pass satellite acquisition geometry, and from the potential soil moisture content during the dry acquisitions, i.e., characterized without precipitation in the days preceding the acquisitions. However, these first limitations are negligible when the radar acquisitions are cautiously selected (see details in the CCD–PCA Method).

The second set of limitations is associated with pre-storm erosion, referring to potential sediment deposition through wind, residual noise from surface roughness, and changes in vegetation coverage. We present a comprehensive discussion and assessment of these pre-storm erosion ambiguities in the Methods and Supplementary Information (refer to "Identification of the intrinsic CCD–PCA error"). These above limitations result in a 13% false-positive error across the whole ROI.

Furthermore, we underline that PCA requires a linearity within the InSAR coherence dataset. Although InSAR coherence decays exponentially over an extended period, our study, constrained to a short period of 72 days, maintains a linear decay. This linear characteristic allows the applicability of PCA within this time frame, thereby ensuring the prerequisite for its reliable use in our CCD–PCA analysis (see "Requirement for effective PCA in the CCD" section in Supplementary Information for details).

## Data availability

All the data used in this study is publicly available online and in open access. The SAR raw images, CCD–PCA products, and ArcGIS package maps used in the manuscript are available at the following repository: https://doi.org/10.17605/OSF.IO/M9C7R. The Copernicus Open Access Hub (https://dataspace.copernicus.eu/) provides access to Sentinel-1 and Sentinel-2 images; daily accumulation precipitation data from the Global Precipitation Measurement (GPM) mission can be accessed through the NASA Giovanni web application (https://giovanni.gsfc.nasa.gov/giovanni/); rainfall accumulation map produced by the Emergency Response Coordination Centre (ERCC) is accessible in their portal (https://erccportal.jrc.ec.europa.eu/ECHO-Products); the watershed extents are available from the HydroATLAS database[73] (https://www.hydrosheds.org/products/hydrobasins).

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

## Acknowledgements
This work is supported by the Arid and Water Research Center (AWARE)
at the University of Southern California (USC) under an award from the
NASA Jet Propulsion Laboratory (AWD#00630) and the USC Zumberge
Research and Innovation Fund awarded to E.H.

## Author contributions
All authors conceptualized the study, interpreted the results, and wrote
the manuscript. J.C.L.N. formulated the methodology, analysed the data
and prepared the figures. Funding was acquired by E.H. Supervision was
provided by E.H.

## Competing interests
The authors declare no competing interests.

## Additional information
**Supplementary information** The online version contains
supplementary material available at

Essam Heggy.

**Peer review information** *Nature Communications* thanks Mahmoud
Mansour, Matthew Wilson, and the other, anonymous, reviewer for their
contribution to the peer review of this work. A peer review file is available.

