## [Peer Review File · Nature Communications]

Assessing Flash Flood Erosion Following Storm Daniel in LibyaReviewers' comments:

Reviewer #1 (Remarks to the Author):

Comments and Suggestions for Authors

Many thanks for the review invitation to the manuscript titled "Storm Daniel's Catastrophic Flood Aftermath in Libya". It is read with great interest and is a well-written manuscript. This study utilized Coherent Change Detection in conjunction with Principal Component Analysis to detect runoff erosion using seven coherence pairs. The collected data included SAR images pre- and post-Daniel's storm for Derna and Susah cities in Libya, in addition to precipitation, topographic data and watershed extent. Flood erosion maps were presented, which are useful for policymaking to address vulnerabilities and avoid probable future damages. However, it is well known in hydrology and hydraulics that areas around ephemeral streams or watershed drainage networks are erosion-prone regions. This paper is interesting from my point of view and my recommendation is to make a major revision before publication in Nature Communications.

Some comments are as follows:

In the introduction section, the knowledge gap should be clearly defined.

Regarding line 94, is it possible to differentiate between sediment deposition and flood erosion using CCD?

In line 373, it is recommended to provide quantitative validation such as the ROC curve as implemented by Wahba et al. (2023), "A Novel Estimation of the Composite Hazard of Landslides and Flash Floods Utilizing an Artificial Intelligence Approach."

For results, comparison with the literature is lacking.

Investigation of associated morphological characteristics to facilitate the mapping of similar regions before disaster occurrences.

For Figure. 5, many words and lines are not clear.

In line 725, it seems that Train and Teros are typos.

Reviewer #2 (Remarks to the Author):

The paper discusses the flooding event originating from Mediane Daniel. In particular, the authors focus on assessing the consequences in terms of soil and shoreline erosion following this extremely damaging event.

The authors used a fairly traditional method based on SAR satellite imagery and the accuracy of the resulting maps seems rather weak: it is calculated by comparing the result of the pre-event analysis with the post-event analysis, without validation from other sources or from other imagery (there are optical images that could provide information on flooded areas. The accuracy in my opinion is therefore unreliable).

The paper is shown to be original because there have not been many studies to date on Mediane Daniel and the flooding it caused. However, I think the article needs considerations at least in the introduction about the consequences of the hurricane in terms of deaths, evacuees and damage caused. There are already a couple of studies in the literature that I found by searching quickly online that I think it is not difficult for the authors to add these numbers to frame the problem.

The authors estimate 9 % and 18 % erosion in the surface area of the cities of Derna and Susah, and this seems to be the only tangible result of this research. The authors repeat this no less than 4 times within the manuscript although it is not clear how this percentage was calculated. I would clarify what numbers these assessments come from.

The results shown by the authors cannot be a meaningful reference until it is clarified how the maps were obtained. It is not clear which pairs of images were considered to draw the maps of flooded areas or how they were combined. The method and supplementary materials are so incomplete that it seems like a maze to get to the bottom of what was done.

The methods and analyses performed need much clarification because under these conditions it is not possible to reproduce the work

1) Which images were selected? Since only a few images are mentioned, it is requested to include a table with Sentinel-1 (reading the dates from Figure 5 is very difficult) and Sentinel-2 images.

2) Explain the method following a step-by-step scheme, starting with SAR image pre-processing and continuing with CCD and PCA techniques in a single discussion. The division of the method

between the main text and supplementary materials is really difficult to follow. Certain concepts are repeated, others are left out.

3) It would be desirable to show flooded areas resulting from single SAR images. Indeed, the resulting erosion maps seems very large and I would like to see the comparison with the single images.

4) Since the 'rapid flood' maps were also produced by Copernicus Emergency Management Service it would be good to see the flooded areas produced by them and make a comparison to indicate and evaluate the real value added by this analysis.

I think there needs to be a substantial revision of the work.

Reviewer #3 (Remarks to the Author):

The paper presents an analysis of erosion caused by flooding from Storm Daniel in Libya in September 2023. This event was very extreme and had a high impact on the area, with several thousand deaths recorded. As such, it will form an important case study, particularly when considering the potential of climate change to increase the frequency of such events. In my view, a comprehensive analysis of the event (i.e. its causal factors, the multifaceted impacts, emergency response, risk management, and changing likelihood under climate change), would be good paper for Nature Communications. However, although the analysis presented is generally sound, the paper as submitted falls short of the level needed.

The primary focus and contribution of the paper is on the mapping of areas of erosion using C-band SAR images, using a Coherent Change Detection method. Much of the paper is devoted to the use of this method, and an explanation of its applicability to other areas, and it would be good to see this analysis published. However, beyond quantifying the area impacted (as a percentage of selected areas of interest, so not directly meaningful), no further analysis of impact is made. Are these areas populated? What is the land use of the areas eroded? How many people were affected? What is the estimated economic impact? Further, no analysis of deposition of eroded sediments was performed, although the impacts of this is likely significant. In addition, while the dam collapses upstream of Derna are mentioned, no analysis specific to this is included (e.g., cause, and erosion/ other damage directly resulting from the collapses). As a consequence of the narrow focus of the analysis, the title of the paper is not appropriate and requires revision.

In its current form, I cannot recommend publication of the manuscript in Nature Communications. I consider that the work is worthy of being published, but it is not at an appropriate level for this journal. Either a substantial reworking is required or, more likely, it should be submitted to a discipline specific journal, such as one in the area of geomorphology or hazards.

Response to Reviewer #1: Reviewer's comments are in black and author's replies are in blue.

We thank Reviewer #1 for his thorough and constructive reviews, comments, and suggestions, which we have fully incorporated in this revised version of the manuscript. We believe this new improved version adds more clarity and addresses all the uncertainties highlighted by Reviewer#1.

Comments and Suggestions for Authors

Many thanks for the review invitation to the manuscript titled "Storm Daniel's Catastrophic Flood Aftermath in Libya". It is read with great interest and is a well-written manuscript. This study utilized Coherent Change Detection in conjunction with Principal Component Analysis to detect runoff erosion using seven coherence pairs. The collected data included SAR images pre- and post-Daniel's storm for Derna and Susah cities in Libya, in addition to precipitation, topographic data and watershed extent. Flood erosion maps were presented, which are useful for policymaking to address vulnerabilities and avoid probable future damages. However, it is well known in hydrology and hydraulics that areas around ephemeral streams or watershed drainage networks are erosion-prone regions. This paper is interesting from my point of view and my recommendation is to make a major revision before publication in Nature Communications.

Some comments are as follows:

Comment 1: In the introduction section, the knowledge gap should be clearly defined.

Accepted: We agree with the Reviewer #1 that the knowledge gap was unclear. The studies conducted on Storm Daniel in Libya have been predominantly centered on coastal cities. However, there has been a notable lack of research on watershed-scale erosion assessment and hydrological dynamics. In our revised manuscript, we can now read lines 83-97 p3:

"As such, characterizing the vulnerability of watersheds and their outlets is essential for effective flood management, avoiding further humanitarian disasters in this instable area. However, only limited mapping of the impacts of Storm Daniel was explored, focusing mainly on the coastal urban areas and neglecting the inland parts constituting the watersheds. For instance, a few days following this catastrophic landfall, Copernicus Emergency Management Service (CEMS) conducted 'rapid flood' mapping from photo-interpretation for the cities of Derna, Benghazi, Al Marj, Al Bayda, Shahat, Tobruk and Susah. They delineated the flood extent and delivered an assessment emergency mapping¹⁶. Then, the European Union, United Nations, and World Bank Group collaborated to publish, in January 2024, a comprehensive report on the catastrophe¹⁰, including 'flood intensity mapping' and the flooding extent. In their report, they emphasize the limited availability of ground truthing data, leading to the use of social media analytics, cell phone data, and night light satellite observations. However, these efforts did not characterize the runoff soil erosion within the watershed, generating dense turbid flows, which modulate the damages of the catastrophic floods"

Comment 2: Regarding line 94, is it possible to differentiate between sediment deposition and flood erosion using CCD?

Accepted: We added a clarification in the introduction that explains that CCD identifies surface changes and without discriminating the deposition from erosion. However, we performed in the revised manuscript a classification to identify sediment loading from CCD and other physical factors.

Lines 107 to 112 p3 in the introduction now reads:

“This CCD approach enables the identification of changes in spatial patterns, such as monitoring flood erosion¹⁸⁻²⁰, displacement of surface scatterers, and sediment deposition in run-off channels²¹, but also to assess different levels of infrastructure damages after a rainstorm^{22,23}. These changes can be attributed to both erosion and sediment load processes. Given the inherent interconnection between the two processes, we frequently use the term “flood surface erosion mapping” in our investigation to designate both processes.”

And from line 112 to 115 p3 we can read:

“However, we can delineate the sediment loading areas by conducting a comprehensive classification. This approach relies on erosion hotspots identified through CCD analysis, hydrological networks, precipitation and topography.”

Comment 3: In line 373, it is recommended to provide quantitative validation such as the ROC curve as implemented by Wahba et al. (2023), “A Novel Estimation of the Composite Hazard of Landslides and Flash Floods Utilizing an Artificial Intelligence Approach.”

Accepted: We acknowledge the importance of quantitative validation in our investigation using ground truth data. Therefore, in our revised version, we integrate a quantitative validation within the “Validation using the CEMS damage assessment map” section in p15-16.

We determine in Lines 608 to 609 p 16 that:

“We find that our CCD–PCA damage map aligns well with the CEMS dataset for both cities, achieving a true-detection rate of up to ~65 % of the surface area.”

Comment 4: For results, comparison with the literature is lacking. Investigation of associated morphological characteristics to facilitate the mapping of similar regions before disaster occurrences.

Accepted: We have included new contextual information in introduction, on the flood damage itself in Lines 56 to 72 p 1:

“The extreme hydroclimatic conditions generated by Storm Daniel caused deluges and mudflows in the storm landfall area (...) For instance, 10 % of the houses were destroyed, and 18.5 % were damaged¹⁰. (...)These figures mark Storm Daniel as the deadliest storm in all the African continent since 1900¹⁰.”

We added a whole section on the “Gained value in using radar observations” to study the flood erosion, in lines 364 to 393 p10:

“Figure 2 shows the before/after map of the flooding in Derna, captured by Sentinel-2 multispectral and Sentinel-1A SAR satellites for the summer and fall of 2023. (...) wet flooded extents could be identified using the Non-Coherence Change Detection (NCCD) technique^{23,59}. NCCD is performed by comparing SAR backscatter amplitudes between acquisitions to identify specular reflections on flooded areas and increases in dielectric on wet soils.(...) Therefore, the NCCD above-mentioned example over Derna and radar

scattering principle underscore the rationale for using SLC SAR images rather than SAR amplitude images to observe flood aftermaths in arid regions.”

Line 355 to 359 p9 now read:

“Additionally, we leverage the resulting flood erosion map to evaluate the erosion hotspots, an important step to estimate sediment loading spatially in wadis. This latter assessment is essential as sediment load intensifies the turbidity of flash floods and influences the likelihood of mudflow occurrences in watershed outlets during future rainstorms.”

We added a comprehensive classification to identify the sediment loading in wadis, which can modulate future flash floods. Lines 200 to 203 p6, read:

“We perform a first-order classification of this run-off erosion and sediment balance dynamics on steep gradients above 3 ° and hillslopes, where loose sediments are washed away and subsequently deposited below in wadis, increasing the likelihood of future mudflow events during rainstorms.”

Comment 5: For Figure. 5, many words and lines are not clear.

Accepted: Figure 5 became Figure 10 in the revised version. We simplified the diagram and improved the resolution. The list of coherence pairs that was previously listed in the figure is now enumerated in Table I.

Figure 10. Coherent change detection (CCD) is associated with the principal component analysis (PCA) method to identify the flood erosion occurring during and after Storm Daniel. The timeline displays the dates of the eight SAR acquisitions used to produce the coherence pairs. The coherence pairs are listed in Table I.

Comment 6: In line 725, it seems that Train and Teros are typos.

Accepted: T_{rain} and T_{eros} are now replaced by H_{rain} and H_{eros} in line 1163 p44

Response to Reviewer #2: Reviewer’s comments are in black and author’s replies are in blue.

We thank Reviewer #2 for his thorough and constructive reviews, comments, and suggestions, which we have fully incorporated in this revised version of the manuscript. We believe this revised improved version adds more clarity and addresses all the uncertainties highlighted by Reviewer#2.

Furthermore, we have made publicly available all the datasets used in this investigation to allow our results reproducibility, including the SAR raw images, CCD–PCA products, and ArcGIS package maps in the following repository: <https://doi.org/10.17605/OSF.IO/M9C7R>

The paper discusses the flooding event originating from Medicane Daniel. In particular, the authors focus on assessing the consequences in terms of soil and shoreline erosion following this extremely damaging event.

Comment 1: The authors used a fairly traditional method based on SAR satellite imagery and the accuracy of the resulting maps seems rather weak: it is calculated by comparing the result of the pre-event analysis with the post-event analysis, without validation from other sources or from other imagery (there are optical images that could provide information on flooded areas. The accuracy in my opinion is therefore unreliable.

Accepted: We agree that the method used is fairly traditional, which minimize technical errors, however its application to this unique case in Libya is the novelty of this paper. As now mentioned in Line 101 to 104 in p3:

“We achieve this objective by utilizing Synthetic Aperture Radar (SAR) satellite scenes to produce interferometric (InSAR) coherence images and applying the well-established Coherent Change Detection (CCD) method, which measures the temporal variations between InSAR coherence images¹⁷. “

Moreover, we agree that the term “accuracy” is too general and leads to misunderstandings depending the context. Therefore, in our revised version, we removed the term “accuracy” and replaced it with precise wording depending the context, such as “omission error”, “commission error” (=“false-detection”), and “intrinsic error”.

For instance, the “CCD-PCA Method” Section now reads in lines 460 to 463 p12, now reads:

“This intrinsic error in the CCD–PCA method accounts for a 13 % occurrence probability (refer to “Identification of the intrinsic CCD–PCA error” in Supplementary Materials for details). This bias identification improves the reliability of our flood erosion mapping process.”

We also agree that the exactitude of the measure needs validation from other sources. Therefore, in our revised manuscript we provide a qualitative and quantitative validation of our CCD method with two distinct datasets. Line 545 to 557 p15 of “Validation of CCD–PCA” section reads:

“We validate our CCD–PCA observations over the cities of Derna and Susah through verification with two distinct sources. The first source was published in the European Union, et al. report and represents the flooding intensity map generated from flow

modeling, derived from a 12-meter resolution Digital Surface Model (DSM). (...).The second source was produced by Copernicus Emergency Management Service ¹⁶ and represents the damage assessment evaluated through photo-interpretation, as depicted in Figure 7.(...)

Furthermore, the “Urban damages in Derna and Susah” Results and discussion section now reads in lines 222-227 p6:

“These observations are validated with two distinct sources, showing a true-detection of 61 – 66 % and 42 – 65 %, a false-detection of 7 – 29 % and 18 – 34 %, and an omission error of 6 – 32 % and 2 – 40 %, for Derna’s city center and Susah, respectively (see “Validation of CCD–PCA” in the Method section for more details). These figures exhibit comparable ranges of value for both cities, reflecting a constant degree of validation of the CCD–PCA method.”

Comment 2: The paper is shown to be original because there have not been many studies to date on Mediane Daniel and the flooding it caused. However, I think the article needs considerations at least in the introduction about the consequences of the hurricane in terms of deaths, evacuees and damage caused. There are already a couple of studies in the literature that I found by searching quickly online that I think it is not difficult for the authors to add these numbers to frame the problem.

Accepted: In our first version we noted “catastrophic death toll” and “deadly flash flood” without giving numbers because the figures were only estimations. At the time of writing, the death toll was estimated to be between 5000 and 20000, then at the time of the submission the figure stood at 11000. In our revised manuscript, we provide the updated numbers as well as the socio-economic context of the region.

In the introduction, we can read now from line 56 to 82 p1-2:

“The extreme hydroclimatic conditions generated by Storm Daniel caused deluges and mudflows in the storm landfall area (see Figure 1) primarily constituted of rangelands (grasses and shrubs) and bare soils ¹¹. Despite the limited urban footprint in the storm landfall area ¹⁰, significant infrastructure damage has been observed, including 5 % of the road network being destroyed, 50 % becoming inaccessible ¹⁰, and the collapse of two flood-control dams upstream of the city of Derna on September 11th ¹² (see Figure 1). The latter triggered a deadly flash flood flowing through the city, located on the river’s deltaic outlet (see Figure 2), destroying a substantial portion of the city buildings, urban infrastructure, and bridges ¹³, which resulted in the accumulation of 8.8 million tons of debris¹⁴. For instance, 10 % of the houses were destroyed, and 18.5 % were damaged ¹⁰. The storm notably affected the city of Derna but also other coastal cities. For example, 28 % of the houses in Susah were destroyed ¹⁰. Albayda, Al-Marj, Shahat, Taknis, Battah, Tolmeita, Bersis, Tokra, and Al-Abyar have also witnessed severe damages, causing a total of 5898 casualties, 8000 missing, 44800 people displaced, and 18838 houses damaged or destroyed in the whole coastal area encompassing Benghazi, Jabal Al Akhdar, Al Marj and Derna ¹⁴ (see Figure 3 and Figure 4). These figures mark Storm Daniel as the deadliest storm in all the African continent since 1900 ¹⁰.

This eastern coast of Libya is one of the main hubs for energy export to Europe. This catastrophe will cost US\$ 1.8 billion in repairing the coastal damages ¹⁰. This figure includes the devastation of crops and harvests across 16209 ha, along with the

destruction of vital topsoil for 4.6 % of the agricultural area ¹⁰. Additionally, 74363 animals perished, accounting for 3.2 % of the region's livestock strategic reserve ¹⁰.

Furthermore, this catastrophe is worsened by pre-existing humanitarian needs and restricted road access due to political instability and intermittent civil wars since 2011 ¹⁵. These factors impede the reconstruction efforts, potentially leading to regional negative economic growth over the coming years and delaying the recovery of the national economy ¹⁰.”

Comment 3: The authors estimate 9 % and 18 % erosion in the surface area of the cities of Derna and Susah, and this seems to be the only tangible result of this research. The authors repeat this no less than 4 times within the manuscript although it is not clear how this percentage was calculated. I would clarify what numbers these assessments come from.

Accepted: These percentages are determined by calculating the ratio of the CCD flood erosion area to the total surface area within the city limits. No elaborate calculation was performed to retrieve these numbers. In the “Urban damages in Derna and Susah” section of Results and Discussion, we can now read in the revised version from line 216 to 222 p6:

“Overall, our CCD–PCA mapping reveals that 18 % of Susah’s surface area exhibits damage and flood surface changes. This percentage is determined by calculating the ratio of the CCD–PCA flood surface change to the total surface area within the city limits. Similarly, in Derna, two watershed outlets present notable damages, erosion and deposits, particularly within the wadis and deltas. Figure 7 shows the extent of damages within the city and its surroundings, derived from CCD–PCA, covering 9 % of its surface area and 27 % of Derna’s city center.”

Moreover, we now provide a damage severity classification by combining our CCD-PCA results with the observations from Copernicus Emergency Management Service. Lines 233 to 245 p6-7 now read:

“Our comprehensive classification now effectively distinguishes between areas of high, moderate, and low damage. The corresponding damage severity maps offer the large-scale snapshot capability of SAR remote-sensing from which the damage magnitude is unclassifiable, with the qualitative capability of CEMS’ photo-interpretation and ground-truthing. The assumptions and criteria to perform this classification are detailed in the “Classifying damages in Derna and Susah” section in Method.

We perform this classification within the flooded extent of Derna’s city center and observe that the damage severity for high, moderate, and low categories accounts for 36.5 %, 29 %, and 27 % of the surface area, respectively, with 7.5 % remaining undamaged (see Figure 5). Moreover, the same classification within the urban area of Susah, as outlined in Figure 6, reveals that 14 % of the surface area exhibits high damages, 34 % displays moderate damages, 39 % shows low damages, and 13 % remains undamaged.”

The classification method used to combine the datasets and calculate these values, is detailed in the “Classifying damages in Derna and Susah” section from Method and in Table II, from line 470 to 508 p13-14.

Comment 4: The results shown by the authors cannot be a meaningful reference until it is clarified how the maps were obtained. It is not clear which pairs of images were considered to draw the maps of flooded areas or how they were combined.

Accepted: The pairs of images considered are indeed crucial and were provided in Figure 10. For better clarity, we improved the revised manuscript by simplifying the Figure 10 and placing the list of the coherence pairs (Table I) in the CCD-PCA Method section. Moreover, we now list the SAR images employed for identification of flooded erosion (Table III) in the “Supplementary materials” section (see table in Comment 6).

The Method section in lines 467 p12 now reads:

Table I. Coherence pairs used in the CCD-PCA analysis.

InSAR coherence pair		CCD-PCA with storm occurrence	CCD-PCA without storm occurrence	Interval (days)	Perpendicular baseline (m)
2023-06-21	2023-08-08	×	×	48	68
2023-07-03	2023-07-27	×	×	24	45
2023-07-03	2023-07-15	×	×	12	50
2023-07-27	2023-08-20	×	×	24	53
2023-07-15	2023-08-20	×	×	36	41
2023-08-20	2023-09-25	×		36	41
2023-07-27	2023-10-07	×		72	23

Line 944-949 p30, read:

“Figure 10. Coherent change detection (CCD) is associated with the principal component analysis (PCA) method to identify the flood erosion occurring during and after Storm Daniel. The timeline displays the dates of the eight SAR acquisitions used to produce the coherence pairs. The coherence pairs are listed in Table I.”

Comment 5: The method and supplementary materials are so incomplete that it seems like a maze to get to the bottom of what was done. The methods and analyses performed need much clarification because under these conditions it is not possible to reproduce the work.

Accepted: Ensuring the reproducibility of our investigation was a priority. We addressed this concern in response to Comment 4 and Comment 5.

Moreover, we reorganized the Method section, which now enclose a sequential description to explain the CCD-PCA Method, two classifications (damage severity classification and sediment loading classification), and CCD-PCA validation. Furthermore, we redesigned the CCD-PCA Method diagram (Figure 10) and placed the information that are non-essential in the supplementary material section.

Additionally, we uploaded the SAR raw images, CCD-PCA products, and ArcGIS package maps in the following repository: <https://doi.org/10.17605/OSF.IO/M9C7R>

Comment 6: 1) Which images were selected? Since only a few images are mentioned, it is requested to include a table with Sentinel-1 (reading the dates from Figure 5 is very difficult) and Sentinel-2 images.

Accepted: We simplified this figure and now listed the InSAR pairs in Table I (see answer to comment 4). Moreover, the SAR images are listed in Table IV, and the Multispectral images are listed in Table V in the “Supplementary materials” in page 34:

List of satellite images used for CCD-PCA processing and visualization

Table IV and Table V, presented below, enumerate the SAR and Multispectral images, respectively, utilized in this study to identify and visualize flood erosion.

Table IV. C-band SAR images used in the CCD-PCA

Date (yyyy-mm-jj)	Satellite name	Direction, Polarization and Level	Relative orbit number	Scene ID	Wavelength
2023-06-21	Sentinel-1A	Desc., VV, L1	7	386783	5.6 cm
2023-07-03	Sentinel-1A	Desc., VV, L1	7	388143	5.6 cm
2023-07-15	Sentinel-1A	Desc., VV, L1	7	389532	5.6 cm
2023-07-27	Sentinel-1A	Desc., VV, L1	7	390906	5.6 cm
2023-08-08	Sentinel-1A	Desc., VV, L1	7	392297	5.6 cm
2023-08-20	Sentinel-1A	Desc., VV, L1	7	393833	5.6 cm
2023-09-25	Sentinel-1A	Desc., VV, L1	7	398421	5.6 cm
2023-10-07	Sentinel-1A	Desc., VV, L1	7	399956	5.6 cm

Table V. C-band SAR and multispectral images used for visualization only

Date (yyyy-mm-jj)	Satellite name	Type and Level	Relative orbit Number & Scene ID	Wavelength
2023-09-13	Sentinel-1A	SAR, L1	7 & 396898	5.6 cm
2023-07-09	Sentinel-2B	Multispectral, L2A	50 & 34SFB 50 & 34SEA 50 & 34SFA 50 & 34SEB	Band 4 (665 nm), Band 3 (560 nm), Band 2 (490 nm)
2023-09-07	Sentinel-2B	Multispectral, L2A	50 & 34SEB	Band 4 (665 nm), Band 3 (560 nm), Band 2 (490 nm)
2023-09-12	Sentinel-2B	Multispectral, L2A	50 & 34SFB 50 & 34SEA 50 & 34SFA 50 & 34SEB	Band 4 (665 nm), Band 3 (560 nm), Band 2 (490 nm)
2023-09-15	Sentinel-2A	Multispectral, L2A	50 & 34SEB	Band 4 (665 nm), Band 3 (560 nm), Band 2 (490 nm)

Comment 7: 2) Explain the method following a step-by-step scheme, starting with SAR image pre-processing and continuing with CCD and PCA techniques in a single discussion. The division of the method between the main text and supplementary materials is really difficult to follow. Certain concepts are repeated, others are left out.

Accepted: While the letter format is not conducive to providing an exhaustive step-by-step method that is already well-documented, we agree with Reviewer#2 that our method section was difficult to follow. In our revised manuscript, we provide now more details in the Method section to offer a step-by-step approach that describe the CCD-PCA methodology (see also answer of Comment 5). Moreover, we only placed the non-essential information in the Supplementary material section. Additionally, we updated the Method diagram to provide more clarity (Figure 10).

Comment 8: 3) It would be desirable to show flooded areas resulting from single SAR images. Indeed, the resulting erosion maps seems very large and I would like to see the comparison with the single images.

Accepted: We created a new Figure 2, representing the areas affected by the flash flood in the city center of Derna, before and after flooding, with single SAR and optical images. This figure illustrates the capability of the SAR in sensing through clouds, particularly after a large rainstorm. Moreover, the figure 2 shows the limitations of the SAR amplitude in mapping the surface changes. In contrast, the CCD approach (i.e., comparison between two SLC SAR images) effectively uses the amplitude and phase information of the SAR to identify these changes.

In particular, we added the section “gained value in using radar observations” in Methods in page 10, to underscore the specificity of the SAR and of the CCD.

We can now read in the revised manuscript from line 370 to 393 p10:

“However, it is challenging to visually assess the building damages, flood surface erosion, sediment deposits, and flooded extent from SAR amplitude images only due to the relatively lower spatial resolution compared to the optical ones. (...)

Therefore, the NCCD above-mentioned example over Derna and radar scattering principle underscore the rationale for using SLC SAR images rather than SAR amplitude images to observe flood aftermaths in arid regions.”

Comment 9: 4) Since the 'rapid flood' maps were also produced by Copernicus Emergency Management Service it would be good to see the flooded areas produced by them and make a comparison to indicate and evaluate the real value added by this analysis.

I think there needs to be a substantial revision of the work.

Accepted: We added in this revised manuscript a comprehensive quantitative validation using the Copernicus Emergency Management Service (CEMS) photo-interpretation map, along with a qualitative validation utilizing a Flood Intensity Model published in January 2024 in an official report jointly delivered by the European Union, United Nations, and World Bank. In the revised manuscript the Figure 7 compares over Derna our CCD-PCA surface change map with the CEMS and Euro-UN-WB maps. Figure 8 makes the same comparisons for the city of Susah.

To quantitatively compare the maps we extracted true-detection, false-detection, and omission error between the CCD-PCA and CEMS datasets. For instance, the new manuscript reads at lines 584 to 586 p16 in “Validation of CCD-PCA” from the Method section:

“(…) we determine that the CCD–PCA map corresponds to the CEMS-Case1 map for 61% of the area (true-detection), with a false-detection of 7 % and an omission error of 32 %.”

Moreover, from these above-mentioned numbers, and recognizing the distinct capabilities of CCD-PCA and photo-interpretation in delineating the extent and degree of damages, we derived a classification to assess the severity of damages. Notably, Figure 5 and Figure 6 show the flood damages for the city center of Derna and Susah with three levels of severity (high, moderate and low).

Our revised manuscript now shows a new section in Method from line 509 to 543 p14-15, entitled “Classifying damages in Derna and Susah”, presenting a comprehensive classification to assess the damage severity in these urban areas. The section “urban damages in Derna and Susah” (from line 215 to 248 p6-7) now encompasses results and discussions on this classification.

For instance, we now can read from line 240 to 245 p7:

“We perform this classification within the flooded extent of Derna’s city center and observe that the damage severity for high, moderate, and low categories accounts for 36.5 %, 29 %, and 27 % of the surface area, respectively, with 7.5 % remaining undamaged (see Figure 5). Moreover, the same classification within the urban area of Susah, as outlined in Figure 6, reveals that 14 % of the surface area exhibits high damages, 34 % displays moderate damages, 39 % shows low damages, and 13 % remains undamaged.”

Response to Reviewer #3: Reviewer's comments are in black and author's replies are in blue.

We thank Reviewer #3 for his thorough and constructive reviews, comments, and suggestions, which we have fully incorporated in this revised version of the manuscript. We believe this revised improved version adds more clarity and addresses all the uncertainties highlighted by Reviewer#3.

The paper presents an analysis of erosion caused by flooding from Storm Daniel in Libya in September 2023. This event was very extreme and had a high impact on the area, with several thousand deaths recorded. As such, it will form an important case study, particularly when considering the potential of climate change to increase the frequency of such events. In my view, a comprehensive analysis of the event (i.e. its causal factors, the multifaceted impacts, emergency response, risk management, and changing likelihood under climate change), would be good paper for Nature Communications. However, although the analysis presented is generally sound, the paper as submitted falls short of the level needed.

Comment 1: The primary focus and contribution of the paper is on the mapping of areas of erosion using C-band SAR images, using a Coherent Change Detection method. Much of the paper is devoted to the use of this method, and an explanation of its applicability to other areas, and it would be good to see this analysis published.

However, beyond quantifying the area impacted (as a percentage of selected areas of interest, so not directly meaningful), no further analysis of impact is made. Are these areas populated? What is the land use of the areas eroded? How many people were affected? What is the estimated economic impact?

Accepted: In our revised manuscript, we now provide a land cover map in Figure 11 that also shows the few populated areas.

Lines 56 to 61 p2 now reads:

“The extreme hydroclimatic conditions generated by Storm Daniel caused deluges and mudflows in the storm landfall area (see Figure 1) primarily constituted of rangelands (grasses and shrubs) and bare soils¹¹. Despite the limited urban footprint in the storm landfall area¹⁰, significant infrastructure damage has been observed, including 5 % of the road network being destroyed, 50 % becoming inaccessible¹⁰, and the collapse of two flood-control dams upstream of the city of Derna on September 11th¹² (see Figure 1).”

The exact land cover proportions are now given in lines 958 to 960 p31:

“Notably, shrublands (37.7 %), grasslands–croplands (10.2 %), and bare areas (50.3 %) account for 98.2 % of the ROI's surface area, while urban areas contribute only to 1.4%.”

Moreover, we now give the proportion of flood erosion per type of land cover. Lines 159 to 163 p5 now reads:

“Moreover, we observe that ~22–26 % of the rangelands (21.6 % for croplands and 25.8 % for shrublands), ~11 % of the bare soils and ~11% of urban areas within the ROI are affected by flood erosion, deposits or damages derived from CCD–PCA (see section “Comparison of the flood erosion/deposits with the land cover” in the Supplementary Material for more details).

Furthermore, in the introduction we give the number of people being affected by the catastrophe and the estimated economic impact. Lines 68 to 77 p1 read:

“(…) causing a total of 5898 casualties, 8000 missing, 44800 people displaced, and 18838 houses damaged or destroyed in the whole coastal area encompassing Benghazi, Jabal Al Akhdar, Al Marj and Derna ¹⁴ (see Figure 3 and Figure 4). These figures mark Storm Daniel as the deadliest storm in all the African continent since 1900 ¹⁰.

This eastern coast of Libya is one of the main hubs for energy export to Europe. This catastrophe will cost US\$ 1.8 billion in repairing the coastal damages ¹⁰. This figure includes the devastation of crops and harvests across 16209 ha, along with the destruction of vital topsoil for 4.6 % of the agricultural area ¹⁰. Additionally, 74363 animals perished, accounting for 3.2 % of the region's livestock strategic reserve ¹⁰.”

Comment 2: Further, no analysis of deposition of eroded sediments was performed, although the impacts of this is likely significant.

Accepted: We added in the revised version, a clarification in the introduction that explains that CCD identifies surface changes and cannot discriminate erosion and deposition. The introduction now reads from line 107 to 115 p3:

“This CCD approach enables the identification of changes in spatial patterns, such as monitoring flood erosion ¹⁸⁻²⁰, displacement of surface scatterers, and sediment deposition in run-off channels ²¹, but also to assess different levels of infrastructure damages after a rainstorm ^{22,23}. These changes can be attributed to both erosion and sediment load processes. Given the inherent interconnection between the two processes, we frequently use the term “flood surface erosion mapping” in our investigation to designate both processes. However, we can delineate the sediment loading areas by conducting a comprehensive classification. This approach relies on erosion hotspots identified through CCD analysis, hydrological networks, precipitation and topography.”

We present the methodology used to assess the sediment loading from line 509 to 535 p14:

“(…)The first variable considered herein is the 3-day rainfall accumulation during Storm Daniel; we chose a conservative estimation threshold of 50 mm, which enables the covering of the whole ROI (see Figure 1). The second variable corresponds to the post-storm flood erosion, which is a metric derived from our CCD–PCA method. We define erosion as significant when more than 5% of the surface area within any 2 km × 2 km bin shows a CCD–PCA signal (i.e., 0.2 km²). The third variable under consideration is the average topographic slope gradient within a 2 km × 2 km bin; we use the threshold angle of 3°, a common gradient threshold employed in arid and semi-arid regions to characterize erosion ^{68,69}. (...) Lastly, after characterizing the erosion hotspots, we assess the areas of sediment loading by intersecting the identified hotspots with the locations of wadis.”

Moreover, we present the results of the sediment loading in wadis from line 203 to 214 p6:

“(…)Notably, Figure 4 indicates erosion hotspots and sediment loading after Storm Daniel, in the North and West parts of the ROI, characterized by hillslopes and shrublands (see Figure 11). Moreover, similar observations are made within Wadi Derna. The watershed of Wadi Derna features an average terrain gradient of 4.9 ° (± 5.5), characterized by exposed bare soil that facilitates overland flow

throughout the area towards the wadi. This dense turbid water flow within the wadi further contributed to the collapses of two dams upstream of Derna city. (...) ”

Comment 3: In addition, while the dam collapses upstream of Derna are mentioned, no analysis specific to this is included (e.g., cause, and erosion/ other damage directly resulting from the collapses).

Accepted: We added a discussion of the causes of the collapses in our revised manuscript. However, rigorous analysis on the causes is still under review by other authors from University of South Carolina and Jackson State University. They submitted a manuscript in Nature Portfolio focusing on the topic. The manuscript is already available in preprint. The manuscript from Nemmem et al., entitles “Rising Waters, Falling Dams: Deciphering the Derna Flood Disaster” and is available through this link:

https://assets.researchsquare.com/files/rs-3809203/v1_covered_e2714159-e53f-4045-9211-2efaac034afa.pdf

From line 148 to 153 p4, we can now read:

“Thus, outlet areas are particularly exposed to the above hydraulic hazards that are often mitigated with a sequence of upstream dams. However, the two dams upstream of Derna have collapsed due to the anomalous hydraulic nature (i.e., flow rate and turbidity) of the floods resulting from Storm Daniel, and lack of maintenance. The extent of the damages triggered by these energetic floods and the associated dam failures are illustrated in Figure 5.”

In Line 200 to 209 p6, we can now read:

“We perform a first-order classification of this run-off erosion and sediment balance dynamics on steep gradients above 3 ° and hillslopes, where loose sediments are washed away and subsequently deposited below in wadis, increasing the likelihood of future mudflow events during rainstorms. Notably, Figure 4 indicates erosion hotspots and sediment loading after Storm Daniel, in the North and West parts of the ROI, characterized by hillslopes and shrublands (see Figure 11). Moreover, similar observations are made within Wadi Derna. The watershed of Wadi Derna features an average terrain gradient of 4.9 ° (± 5.5), characterized by exposed bare soil that facilitates overland flow throughout the area towards the wadi. This dense turbid water flow within the wadi further contributed to the collapses of two dams upstream of Derna city.”

From line 311 to 314 p8, we can now read:

“Gravity dams, like those that collapsed in Derna, play an essential role in flood management in arid regions. These structures are designed to withstand rapid changes in hydrostatic pressure caused by changes in environmental load conditions, such as successive droughts and flash floods that can result in deformation and cracking⁵⁵. (...)”

Comment 4: As a consequence of the narrow focus of the analysis, the title of the paper is not appropriate and requires revision.

Accepted: In our revised version, the new title now reads:

“Assessing Flash Flood Erosion Following Storm Daniel in Libya”

REVIEWERS' COMMENTS

Reviewer #1 (Remarks to the Author):

I am genuinely appreciative of the authors' diligent efforts in addressing the comments provided. It is evident that they have thoroughly considered and clarified the points raised, resulting in a manuscript that is now suitable for publication in Nature Communications.

Reviewer #2 (Remarks to the Author):

I thank the authors for the thorough and complete revision of the paper. They edited the document a lot also based on my comments and now the study is complete and I think it deserves to be published. I really liked the new form and the descriptions are now very clear.

Reviewer #3 (Remarks to the Author):

The paper has been substantially revised from the earlier version, and I consider that all concerns raised have been addressed adequately. I suggest some improvements are needed to the figures to bring them up to a good standard:

Figure 1:

- No need to show the red box (just crop the figure to the region of interest)
- The DEM colours are not colour blind friendly, and although continuous colour bar is used, the shading is discrete. I suggest using a simpler shade (e.g. greyscale), with hill shading included.

Figures 5-8, 12-13:

- These would be clearer if a map basemap were used, rather than an image.

Otherwise, I consider that the paper is worth of publication and congratulate the authors for their good work.

ANSWERS (in blue) TO REVIEWER#3 COMMENTS (in black)

Reviewer #3 (Remarks to the Author):

The paper has been substantially revised from the earlier version, and I consider that all concerns raised have been addressed adequately. I suggest some improvements are needed to the figures to bring them up to a good standard:

Figure 1:

- No need to show the red box (just crop the figure to the region of interest)

Accepted: We removed the red box in the revised manuscript.

- The DEM colours are not colour blind friendly, and although continuous colour bar is used, the shading is discrete. I suggest using a simpler shade (e.g. greyscale), with hill shading included.

Accepted: We changed the DEM colors in the revised manuscript. However, we did not apply hillshading, as it amplifies the DEM imperfections in the gently sloping areas.

Figures 5-8, 12-13:

- These would be clearer if a map basemap were used, rather than an image.:

Declined: Our study focuses on the dynamics of soil physical properties. By overlaying the identification of damages and erosion with a true-color optical map, we can better understand the geomorphological context of the area.

Otherwise, I consider that the paper is worth of publication and congratulate the authors for their good work.